# SANTO: a coarse-to-fine alignment and stitching method for spatial omics

Haoyang Li[1,2,5], Yingxin Lin [3,5], Wenjia He[1,2], Wenkai Han[1,2], Xiaopeng Xu [1,2], Chencheng Xu[1,2], Elva Gao[4], Hongyu Zhao [3] ✉ & Xin Gao [1,2] ✉

With the flourishing of spatial omics technologies, alignment and stitching of slices becomes indispensable to decipher a holistic view of 3D molecular profile. However, existing alignment and stitching methods are unpractical to process large-scale and image-based spatial omics dataset due to extreme time consumption and unsatisfactory accuracy. Here we propose SANTO, a coarse-to-fine method targeting alignment and stitching tasks for spatial omics. SANTO firstly rapidly supplies reasonable spatial positions of two slices and identifies the overlap region. Then, SANTO refines the positions of two slices by considering spatial and omics patterns. Comprehensive experiments demonstrate the superior performance of SANTO over existing methods. Specifically, SANTO stitches cross-platform slices for breast cancer samples, enabling integration of complementary features to synergistically explore tumor microenvironment. SANTO is then applied to 3D-to-3D spatiotemporal alignment to study development of mouse embryo. Furthermore, SANTO enables cross-modality alignment of spatial transcriptomic and epigenomic data to understand complementary interactions.

Spatial omics, profiling molecular characteristics with spatial context on intact tissues, empower the understanding of cellular organization and interaction that govern physiology and pathogenesis[1–3]. Generally, most spatial omics are measured on 2-dimensional (2D) tissue slices, but molecular dynamics and intercellular actions in the tissues can be rarely embodied on them, which is critical information in understanding developmental biology and heterogenous tumor microenvironment (TME)[1,4,5]. Aligning spatial omics data from serial 2D sections to construct a three-dimensional (3D) profile of the entire tissues facilitates a bird's-eye view to enrich a more comprehensive understanding of the tissue ecosystem[4,6–8]. For example, a study applied stereo-seq to late-stage embryos and all stages of larvae, generating multiple serial 2D slices for each timepoint, which were spatially aligned and reconstructed to individual 3D point-cloud-based models[9]. These models were used to detect the functional subregions

in embryonic and larval midgut, analyze spatial cell state changes during larval spermatogenesis and identify active transcription factor regulons. However, 2D spatial omics data without 3D reconstruction cannot supply molecular information over and under the slice, which impedes the study of 3D cell-cell communications in spatial context, even development of organogenesis.

Alignment of slices usually requires that adjacent or replicated slices are fully overlapped in the same region of the tissue, which have similar omics and spatial patterns[6]. In the real-world experiments of spatial omics, regarding the technical difficulties of tissue dissection and array placement, it is rare for two slices to profile identical assayed regions[10]. Therefore, even tiny displacement between slices would be aligned unrealistically by existing alignment methods, resulting in biased molecular characteristics of 3D context. On the other hand, through the explosion of spatial omics data recently, biologists and

[1]Computer Science Program, Computer, Electrical and Mathematical Sciences and Engineering Division, King Abdullah University of Science and Technology (KAUST), Thuwal 23955-6900, Saudi Arabia. [2]Center of Excellence on Smart Health, King Abdullah University of Science and Technology (KAUST), Thuwal 23955-6900, Saudi Arabia. [3]Department of Biostatistics, Yale University, New Haven, CT, USA. [4]The KAUST school, King Abdullah University of Science and Technology (KAUST), Thuwal 23955-6900, Saudi Arabia. [5]These authors contributed equally: Haoyang Li, Yingxin Lin. ✉e-mail: hongyu.zhao@yale.edu; xin.gao@kaust.edu.sa

clinicians tend to investigate cellular organization in the larger and unabridged slices dissected from huge tissues of mammalian species or TME[7,11]. But current technologies can only achieve the capture area up to 15 cm², which hinders the investigation of the larger and unabridged slices dissected from huge tissues of mammalian species or TME[11]. For instance, according to the report of breast cancer staging, tumor would be >5 cm at T3 stage[12,13].

The aforementioned issues can be both solved by stitching tissue slices, conceptually similar to image stitching, which aims to stitch multiple partially overlapped tissue slices together to generate a larger slice[14]. From the problem formulation point of view, the stitching task can be considered a generalization of the alignment task without affection of technical displacement[15]. It aids in the mutual information enrichment between two slices and even allows for complementary spatial resolutions and genomic coverages from cross-platform data[15,16]. More importantly, it also overcomes the constraints of limited capture area imposed by existing technologies, enabling the investigation of sizable slices and integrated ecosystems[10].

Currently, several existing methods, e.g. PASTE and PASTE2, designed for alignment and stitching tasks could not handle slices with unevenly distributed spots, e.g., image-based dataset, or slices generated from cross-platform technologies[6,10]. They are also highly time-consuming and hinder their applicability to large-scale datasets with high-resolution spots. SLAT and STAligner focus on one-to-one mapping instead of plane-to-plane global alignment between two slices[17,18]. STalign requires the manual landmarks of two slices from users for affine alignment and focuses more on the local distortion through alignment[19]. SPACEL needs additional spatial domain annotation of each spot of two slices to aid alignment[20]. In addition, none of these methods can target global alignment and stitching tasks together while effectively addressing the aforementioned issues.

Here we introduce SANTO (coarSe-to-fine AligNment and sTitching for spatial Omics), a coarse-to-fine method targeting alignment and stitching tasks for spatial omics data (Fig. 1A). SANTO meets the challenges of spatial alignment and stitching through the following key innovations: (1) in the coarse procedure, SANTO rapidly identifies

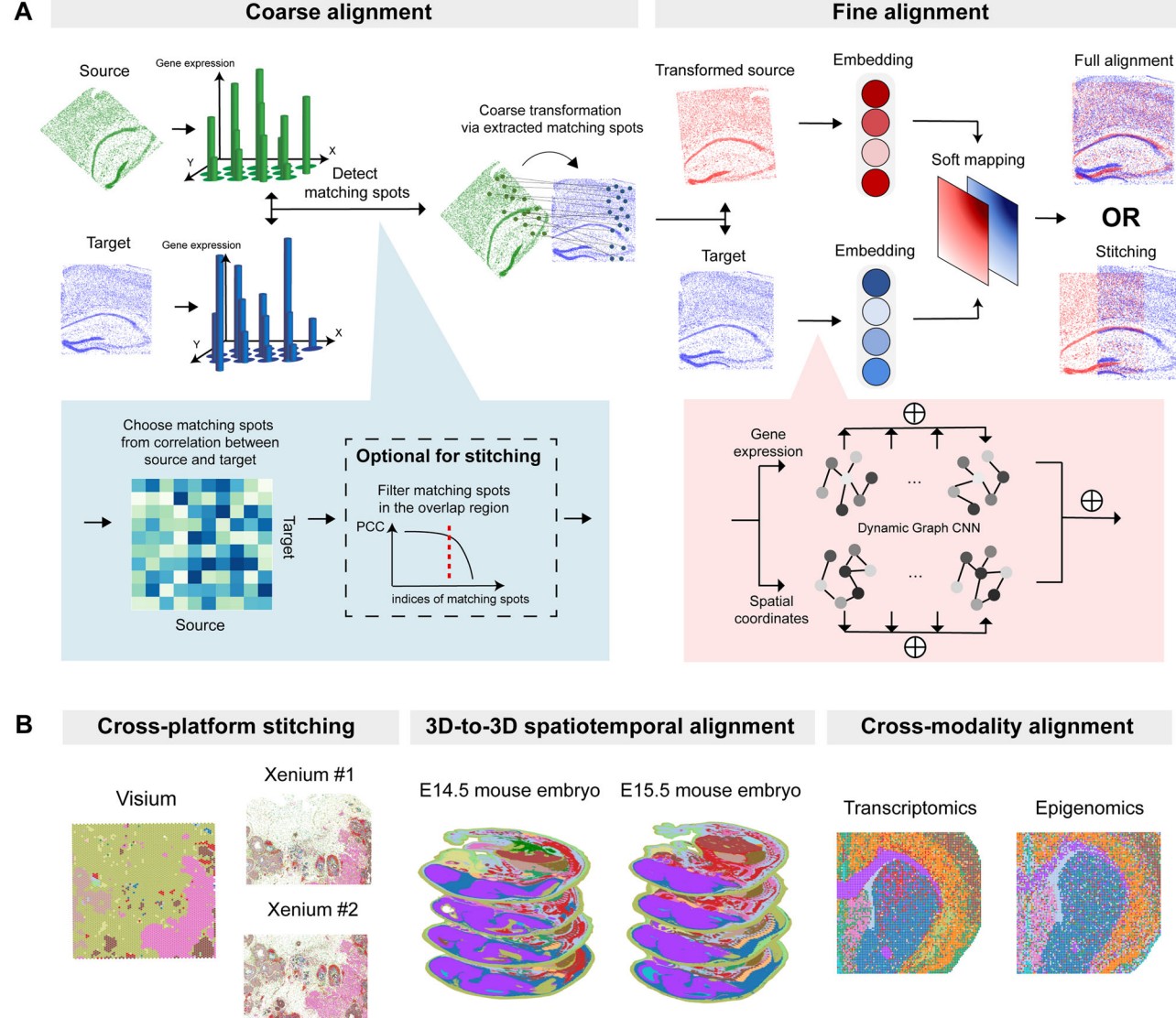

**Fig. 1 | Overview of SANTO. A** Pipeline of SANTO. Firstly, SANTO uses two omics feature expression profiles to identify matching spots for optimizing proper transformation, which aims to quickly find a reasonable initial position for two slices. Optionally, overlap area could be identified by filtering matching spots for the stitching task. In the fine alignment, SANTO utilizes dynamic graph CNN (DGCNN) to dynamically extract the local and global embeddings for spatial coordinates and omics feature expressions. Then, embeddings from two slices are used to generate soft mapping for optimizing transformation. **B** SANTO can be applied to three scenarios in this work: cross-platform stitching, 3D-to-3D spatiotemporal alignment, and cross-modality alignment.

initial spatial positions of two slices and automatically finds the overlapping region between them; (2) during the fine procedure, SANTO refines the positions of slices by considering both omics and spatial patterns locally and globally through dynamic graph updates. Other than the superior performance of benchmarking with existing methods, we demonstrate SANTO's performance through various important but challenging alignment and stitching tasks (Fig. 1B). In particular, using two slices of breast cancer samples from 10x Xenium and Visium, we highlight the ability of SANTO to stitch slices from different spatial platforms with supercellular and subcellular resolutions. Leveraging information from technologies with different genomic coverages and spot resolutions, SANTO enables a series of downstream analysis, including novel cell type identification, prediction of undetected genes' expressions and cell-cell communication of TME. We then apply SANTO, for the first time, to 3D-to-3D spatiotemporal alignment to study the tissue development using mouse embryonic samples from two developmental time points. We further illustrate SANTO's capacity in cross-modality alignment of spatial transcriptomic and epigenomic mouse brain samples to understand complementary interactions between different omics data.

## Results

### SANTO enables alignment and stitching for spatial omics

SANTO formulates spatial omics data in point-cloud-based data structure, enabling alignment and stitching at the 2D or 3D level[21]. To enhance the efficiency of SANTO and prevent alignment from spurious local optimum, we divide the pipeline as a two-step procedure, a coarse alignment followed by a fine alignment. In the coarse alignment, SANTO tends to identify a reasonable initial position of the source slice to the target slice rapidly. SANTO utilizes feature expression profiles from two slices to identify the one-to-one matching spots according to the spot similarity and then extracts the paired spots located inside the overlap region if two slices are expected to be stitched together. According to these matching spots, the source slice would be coarsely and rapidly transformed to the target slice via proper transformation from a singular value decomposition (SVD)-based optimization (Methods).

In the fine alignment, after the constructions of local neighbor graphs for spatial coordinates and omics expressions of spots separately on two slices, SANTO learns permutation-invariant embeddings of the two graphs by dynamic graph CNN (DGCNN)[22,23]. DGCNN allows the graph to update dynamically after each layer in the deep learning network instead of traditionally fixed graphs, according to the difference between proximities in the graphs of embedding and original input. To learn global and local features, SANTO integrates the learned multi-scale embeddings together in the omics and spatial graphs for individual slices. Then, SANTO combines the embeddings from two graphs of each slice and uses two final embeddings from two slices to generate a probabilistic map revealing the combinatorial matching of all spots in the source to target slices, which would help identify final fine-grained transformation. To optimize the transformation, we design the loss function by co-considering spatial coordinates and omics expression levels of spots, under the assumption that proximal spots tend to be similar at spatial and omics level (Methods).

### SANTO outperforms existing methods on diverse datasets and conditions with robust performance

To benchmark SANTO comprehensively, we compare SANTO with existing methods designed for alignment and stitching tasks separately (PASTE[6], SLAT[17], STAligner[18] and PASTE2[10]), in terms of accuracy, robustness and usability, using three datasets generated by STARmap PLUS, Visium and MERFISH technologies which are diverse on spot resolution and sequencing depth[24–26].

Firstly, we focus on the alignment task by comparing SANTO with PASTE, SLAT and STAligner on the STARmap PLUS dataset, including two originally aligned slices from mouse brains with the ages of 8 and 13 months separately[26]. For each pair of slices, we manually rotate one slice with 6 different angles from 45° – 270° and test methods' ability to find the correct alignment. As shown in Fig. 2A, SANTO perfectly aligns slices under large-angle rotations whereas PASTE cannot. We also utilize three metrics to quantify the results, by Pearson Correlation Coefficient (PCC), Cell type matching Index (CI) and Alignment Angle Score (AAS) (Methods). PCC and CI aim to measure the similarity of omics expression level and cell type annotation between spatially proximal spots from two slices. AAS quantifies the accuracy of predicted rotated angle according to the ground truth, where a higher value represents better prediction. Using these quantification metrics, we demonstrate the superior performance of SANTO against PASTE, SLAT and STAligner (Supplementary Fig. 1). Regarding to CI, SANTO (0.35 on average) performs roughly four times better than PASTE (0.09 on average) under six different conditions (Fig. 2B). Notably, using AAS, we find that the predicted angles from SANTO are much closer to the ground truth than PASTE, SLAT and STAligner (Fig. 2B, Supplementary Fig. 3).

To evaluate the accuracy on the stitching task, we compare SANTO with PASTE2, STAligner and SLAT on the Visium dataset[25]. To create the scenarios when two slices to be aligned are only partially overlapped, we manually rotate one of the two original aligned pairs of slices with different angles (30°, 60° and 90°) and then clipping to different percentages of overlap areas (40%, 60% and 80%). Similar to previous benchmarking, we evaluate the alignment performance by visualization and quantification of PCC, CI and AAS from all nine settings. We find that SANTO is generally more accurate in terms of PCC, CI and AAS than PASTE2, SLAT and STAligner in all the settings regardless of rotations and percentages of overlap regions (Fig. 2C, D, Supplementary Figs. 2, 3). We note that unlike PASTE2 that requires users to manually select the percentage of overlap region from 0 to 1, SANTO can automatically and accurately identify the overlapping region without human intervention.

In comparison to alignment of two slices, stacked alignment with multiple serial slices needs to be more precise for the 3D reconstruction of spatial omics. Following the order of given slices, alignment of each paired slices is based on alignment result of previous pairs, which would finally amplify the bias of all slices. To further evaluate the performance on stacked alignment, we consider two MERFISH datasets with 24 consecutive slices from two mouse samples[24]. SANTO, PASTE, SLAT and STAligner aligned each pair of slices according to previous alignment results following the order of slices. By visualizing 2D projections from 3D reconstructions of two samples, only SANTO can reconstruct similar spatial distributions of cell types in the z-axis (Fig. 2E, Supplementary Fig. 7A). We also randomly rotate these slices from 0° to 45° and the results lead to the same conclusion above (Supplementary Fig. 8). PASTE fails to align each pair of slices as it generates a chaotic distribution of cell types on both datasets, and SLAT and STAligner also cannot reconstruct spatial patterns of cell types well with biased transformations (Fig. 2E, Supplementary Fig. 7A). These observations are further confirmed by the quantification metrics PCC and CI, where SANTO is 12% on average higher than PASTE and 15% on average higher than SLAT an STAligner on both samples (Fig. 2F, Supplementary Fig. 7B).

Finally, we demonstrate the robustness and usability of SANTO through three sensitivity analysis: (1) using different hyperparameter settings; (2) selection of target slides; and (3) computational time. Firstly, we examine the alignment performance of SANTO by varying four key hyperparameters, including learning rate, alpha (the weight of omics expression loss), number of epochs and k (number of neighbors during graph construction), using all three benchmarking datasets (Fig. 2G, Supplementary Fig. 10, Methods). We find that SANTO maintains stable performance on different datasets and parameter settings where the most extreme deviation is only around 4%

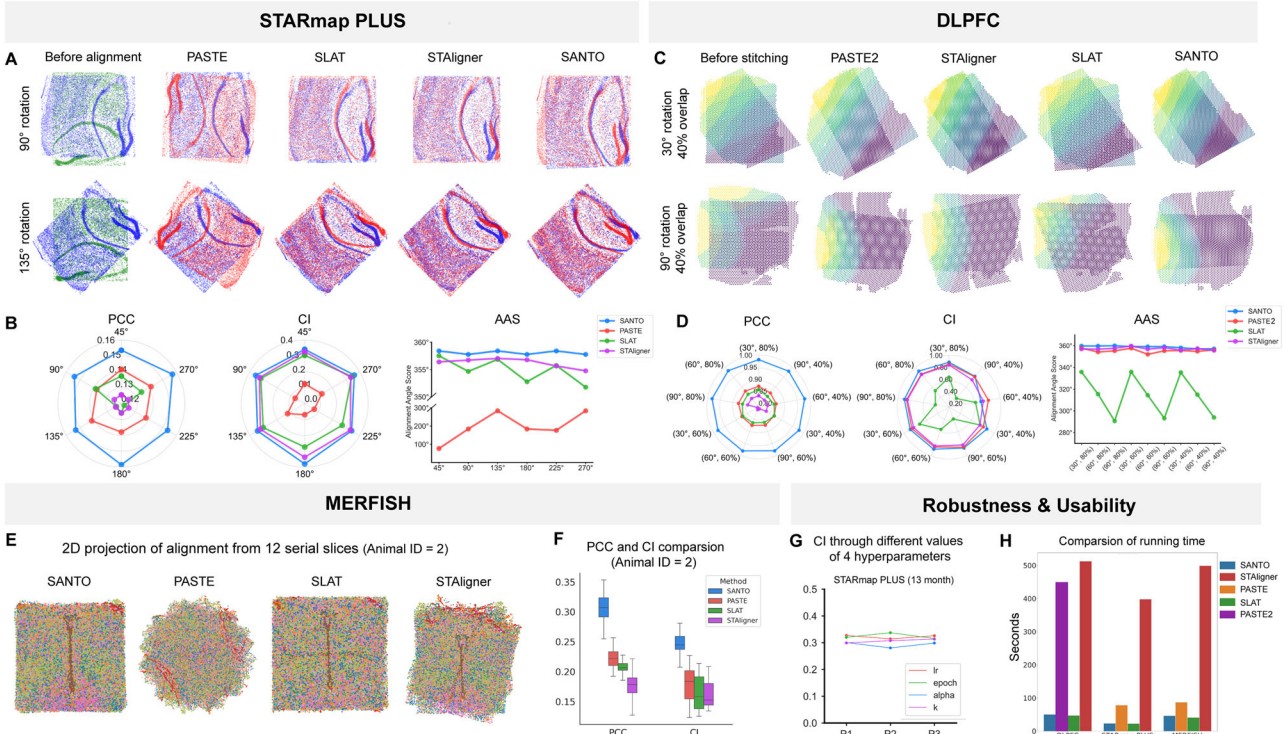

**Fig. 2 | Benchmarking results of SANTO. A** SANTO is benchmarked with PASTE, SLAT and STAligner on the STARmap PLUS dataset for the alignment task. Here are two visualizations of before alignment, results of PASTE, SLAT, STAligner and SANTO in two rotations. Green slices are the source slices before alignment and blue slices mean the target slices. And the red slices are the transformed source slices outputted from different methods. **B** SANTO outperforms PASTE, SLAT and STAligner based on multiple rotations and evaluated by PCC, CI and AAS. **C** SANTO is benchmarked with PASTE2 on the Visium dataset for stitching task. Here are two visualizations of before alignment, results of PASTE2 and SANTO in two rotations and percentages of overlap area. **D** SANTO outperforms PASTE2 with respect to multiple rotations and percentages of overlap area, and evaluated by PCC, CI and AAS as well. **E** To evaluate the performance on multiple serial spatial omics data, SANTO is compared with PASTE, SLAT and STAligner by showing the 2D projections after 3D alignment. **F** PCC and CI among 12 slices are evaluated for PASTE (PCC: 0.22 ± 0.01, CI: 0.18 ± 0.03), SLAT (PCC: 0.20 ± 0.02, CI: 0.16 ± 0.03), STAligner (PCC: 0.17 ± 0.03, CI: 0.16 ± 0.03) and SANTO (PCC: 0.30 ± 0.03, CI: 0.24 ± 0.03), and the results reveal superior performance of SANTO. **G** Robustness is tested of SANTO for different hyperparameters. **H** Compared with PASTE, SLAT, STAligner and PASTE2, the running time of SANTO is shorter in different datasets. Source data are provided as a Source Data file.

(Supplementary Fig. 11). Next, we perform the sensitivity analysis to investigate the effect of the choices of different target slices using MERFISH datasets with multiple slices. We consider first and last slices as target slices respectively to align the rest of the slices ascendingly and descendingly. We find that the selection of target slides has negligible impact on the performance of SANTO, where the averaged difference of PCC and CI is only limited to 0.03 (Supplementary Fig. 7C). Finally, we compare the computational efficiency of SANTO, PASTE, SLAT, STAligner and PASTE2 by comparing their running time. Especially, SANTO is averagely ten times faster than STAligner through all of datasets and eight times faster than PASTE2 in DLPFC dataset. Additionally, we also apply SANTO to various scenarios to guarantee its robustness including adding different noises for gene expression profiles, different number of features for each spot/cell (Supplementary Figs. 4 and 9). The performance from both of experiments proves that SANTO is robust among different noises of gene expressions and number of genes. To test the ability on spatial domain identification task, SANTO is also applied to DLPFC dataset. The visualization results and ARI score reflect the ability of SANTO to distinguish different spatial domains based on the aligned embedding space, although spatial domain identification is not our main goal (Supplementary Fig. 13).

### Alignment of cross-platform slices enables exploring TME with complementary features

To illustrate SANTO's performance in cross-platform alignment and stitching, we obtain three cross-platform datasets from human breast cancer including two Xenium slices and one Visium slice in the same tissue section[27]. Xenium is image-based technology with subcellular resolution and in this case, two Xenium slices profiled 103,209 and 75,095 cells with 313 genes and two Xenium slices were partially overlapped (Fig. 3A). In contrast, Visium, a spot-based technology, profiled 4992 spots with 17,943 genes with super-cellular resolution. Based on the original cell-type annotations, Xenium slices provide more fine-grained cell type annotation than the Visium slice. We first use SANTO to stitch two Xenium slices together to enlarge the view of the Xenium slices where cell types are precisely aligned through visualization (Fig. 3B). Then, Visium and Xenium slices are stitched by SANTO, and we map our stitching result on the original H&E image (Fig. 3A). The consistent spatial patterns across H&E image, Visium and Xenium slices of two tumor regions confirm the ideal alignment between Visium and Xenium slices (Fig. 3C).

Through cross-platform alignment, we identify a more diverse cell type composition on the Visium slice by harnessing the finer resolution offered by the aligned Xenium slices. To achieve this, we deconvolute cell types of spots on the Visium slice based on the dominate cell type from adjacent cells of Xenium within the overlapping regions of the Xenium and Visium slices (Fig. 3D, E) (Methods). We find that deconvolution informed by Xenium could unveil multiple cell types that remains undetected by the previous deconvolution method which only uses scRNA-seq as the reference. This is particularly evident in the immune cell population surrounding tumor cells, including CD4 T cells, CD8 T cells and CD163 macrophages (Fig. 3E). From the markers' expressions of invasive

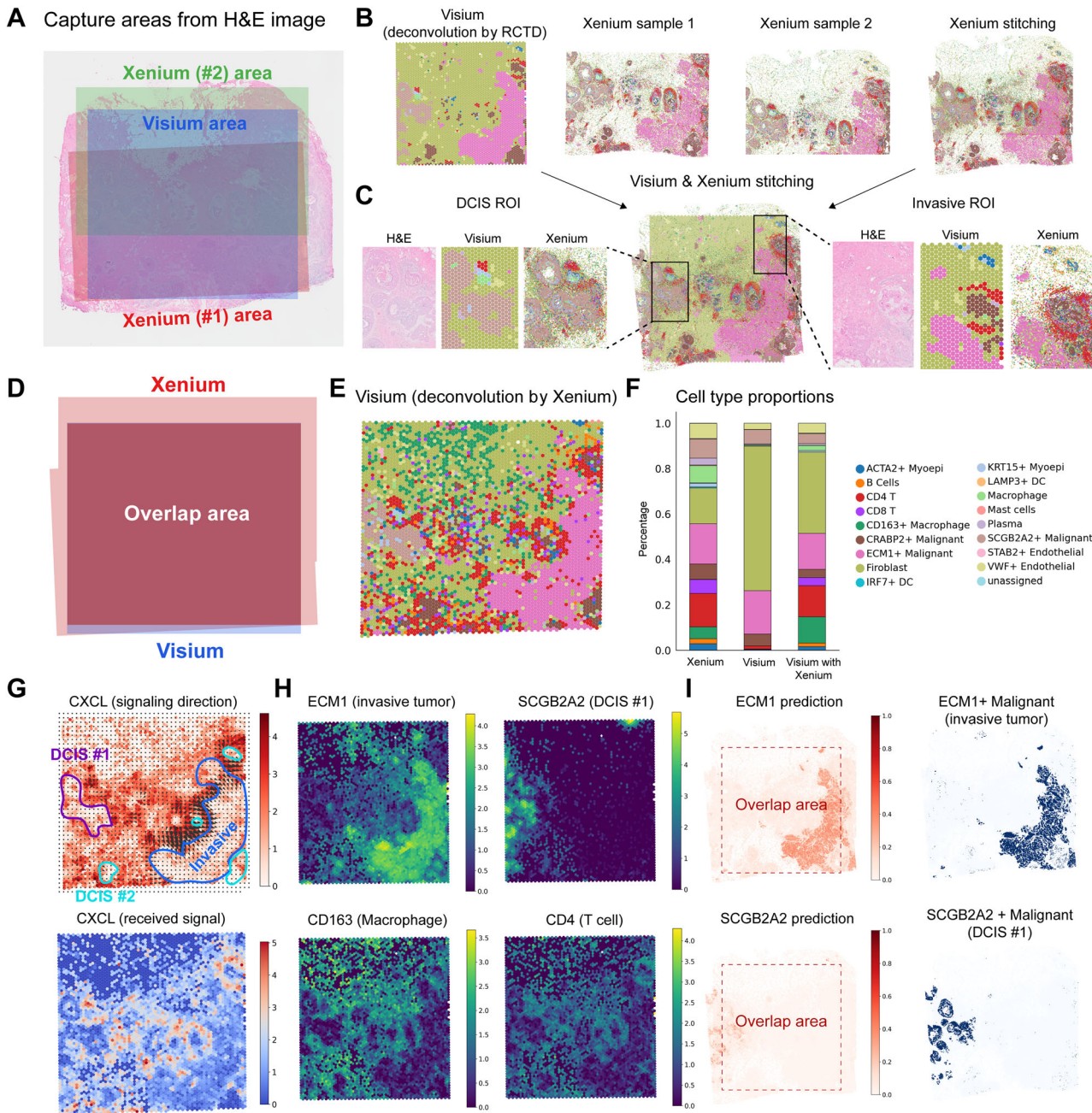

**Fig. 3 | Cross-platform alignment by SANTO. A** The spatial positions of Visium and two Xenium slices after stitching mapping on the H&E image. **B** The visualization of Visium and two Xenium slices, and stitching results of the two Xenium slices. **C** The stitching results between Xenium and Visium. Exact matching between morphological patterns from the H&E image and cell-type distributions from two slices on the two tumor regions reveals the accuracy of SANTO. **D** The overlap region between Visium and Xenium slices based on stitching. **E** New deconvolution results of the Visium slice by accurately aligned Xenium slice. **F** The cell type proportions of Xenium, Visium deconvolved by RCTD and Visium deconvolved by aligned Xenium. **G** Signaling direction and received signal both reveal that CXCL mainly occurs on the boundaries of all tumors. **H** Four markers' expressions in the Visium dataset. Expressions of CD163 and CD4 reveal the reasonability of new deconvolution by Xenium for Visium. **I** The undetected genes ECM1 and SCGB2A2 in Xenium can be predicted by Visium and Xenium stitching. Expressions of two markers outside the overlap area are accurately matched with their cell type distributions. Source data are provided as a Source Data file.

tumor and ductal carcinoma in situ (DCIS) #1, Visium slice deconvoluted by scRNA-seq and Xenium slice both show distinct patterns. However, from the markers' expressions of CD163 macrophages and CD4 T cells, deconvolution by Xenium could detect the corresponding spatial patterns whereas deconvolution by scRNA-seq could not (Fig. 3H). The cell type proportions of the Visium slice deconvoluted by Xenium also reveals that ideal cross-platform alignment supplies novel cell types for the original Visium slice (Fig. 3F).

The enhanced and more precise estimation of the immune cell population in the deconvoluted cell type compositions enable us to investigate the spatial cell-cell communication within the TME. Using COMMOT[28], we focus on the cell-cell communication of the signaling pathway CXCL, which plays a crucial regulatory role in TME that could attract immune cells into tumor tissue which is important for triggering immune responses, regulating inflammation, and promoting antitumor immunity[29,30]. We validate our identification of immune cell populations by confirming their spatial pattern alignment with CXCL

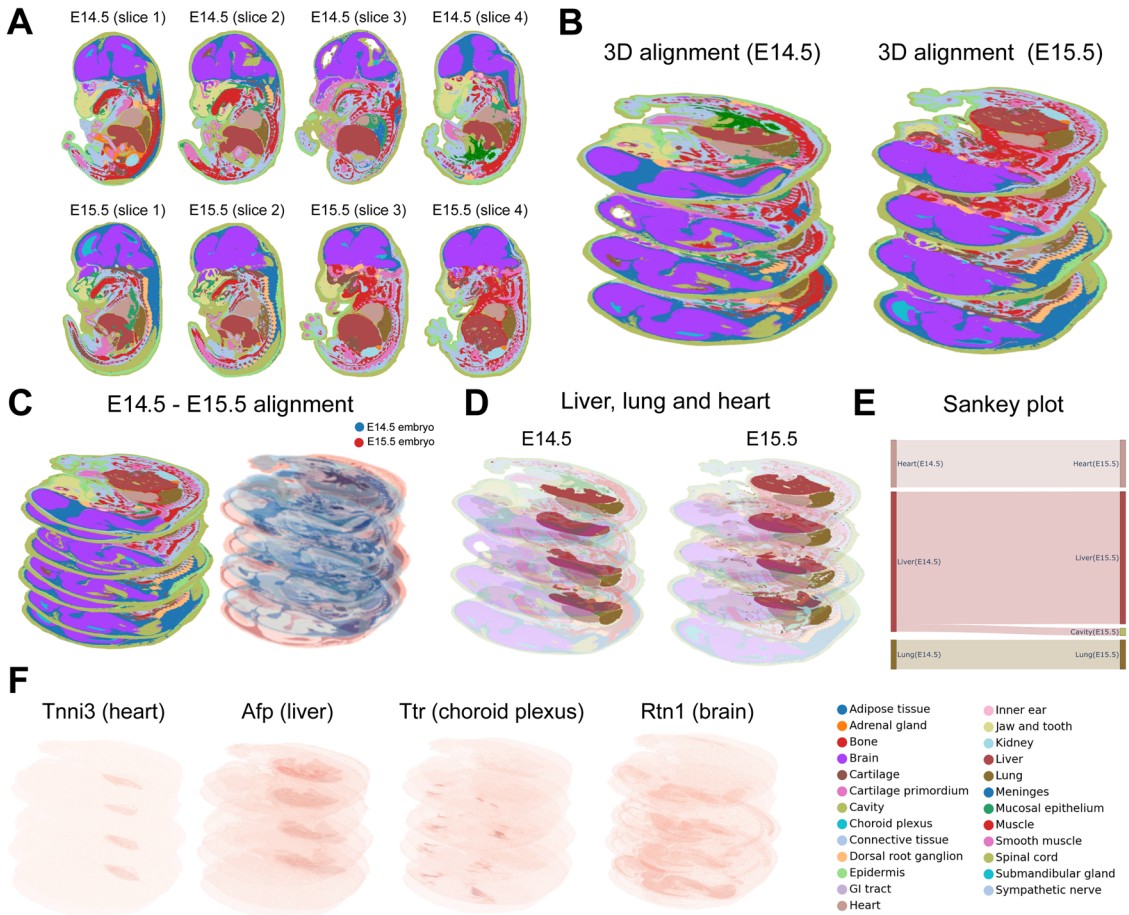

**Fig. 4 | 3D-to-3D spatiotemporal alignment by SANTO. A** Eight samples from E14.5 and E15.5 time points of mouse embryos. **B** 3D alignment of E14.5 and E15.5 embryos by SANTO. **C** 3D-to-3D spatiotemporal alignment of E14.5 and E15.5 embryos. **D** Visualization of liver, lung and heart under 3D-to-3D spatiotemporal alignment. **E** The sankey plot of liver, lung and heart between E14.5 and E15.5. **F** Expressions of several markers in the spatiotemporal alignment results. Source data are provided as a Source Data file.

signaling. The estimated signaling direction and received signal highlight the spatial pattern of CXCL from the ligands CXCL9, CXCL10 and CXCL12 to receptors ACKR1 and CXCR4 around three kinds of tumors: DCIS #1, #2 and invasive tumor (Fig. 3G). The signals of ligand are relatively strong around the invasive tumor that the signaling directions of ligand point inside the invasive tumor. Meanwhile, strong spatial patterns of receptor CXCR4 around DCIS #1, #2 and invasive tumor also provide strong evidence of CXCL's existence.

Complementarily, we could also leverage transcriptome-wide profile of Visium to impute untargeted genes in Xenium. We first build a Poisson regression model to fit 313 genes from Xenium cells with the expression of one gene in the closet Visium spot that we want to detect in the Xenium data within the overlapping region between Xenium and Visium slices. We then apply this model to predict the expression of specific gene for all of Xenium cells beyond the overlapping area (Methods). We focused on the prediction of the expression levels of biomarkers from invasive tumor and DCIS #1 (ECM1 and SCGB2A2) whose distributions across the boundary of the overlapping area. We find that the predicted biomarker expression shares the similar spatial patterns with the cell type annotation, demonstrating the power of using Visium to expand Xenium's expression profiling via alignment and stitching of SANTO.

### 3D-to-3D spatiotemporal alignment helps identify tissue development

We next perform a more challenging task of spatial alignment where we extend SANTO to 3D-to-3D spatiotemporal alignment, using two mouse embryonic samples with four consecutive slices from embryonic day 14.5 and day 15.5 (E14.5 and E15.5) as an example. Here, we use SANTO to align the four slices in each sample into a 3D-level embryonic sample (Fig. 4A, B)[11]. Different from the 2D-to-2D alignment, we extend the dimension of the transformation matrix to 3×3 during SVD-based optimizations after coarse and fine alignment, and SANTO could be easily extended to 3D-to-3D alignment. Thus, we enable 3D-to-3D alignment across two 3D-level embryonic samples from two time points and visualize the distributions of samples and tissues (Fig. 4C). Through the alignment, morphology of most of tissues from two timepoints was similar that they transited smoothly along the tissue development. We extract three main tissues including liver, lung and heart to study the developmental process. We visualize 3D-level aligned distributions of three tissues from E14.5 and E15.5, and these tissues were spatially adjacent to each other (Supplementary Fig. 14). To better visualize the alignment of E14.5 and E15.5, we move aligned E14.5 sample horizontally outside the E15.5 (Fig. 4D). Using Sankey plot to visualize the relationship between the closest pairs of cells from two timepoints, we demonstrate that most of cells at E14.5 transited to the same tissues at E15.5 (Fig. 4E). The continuous patterns of the marker genes of heart, liver, choroid plexus and brain (Tnni3, Afp, Ttr and Rtn1) on the 3D-to-3D alignment result further validate the accurate spatiotemporal alignment through two timepoints (Fig. 4F). Additionally, we also benchmarked SANTO with other methods including PASTE, SLAT and STAligner, which were used to align two samples separately and conduct 3D-to-3D alignment between samples from two timepoints (Supplementary Fig. 14). SANTO could perfectly

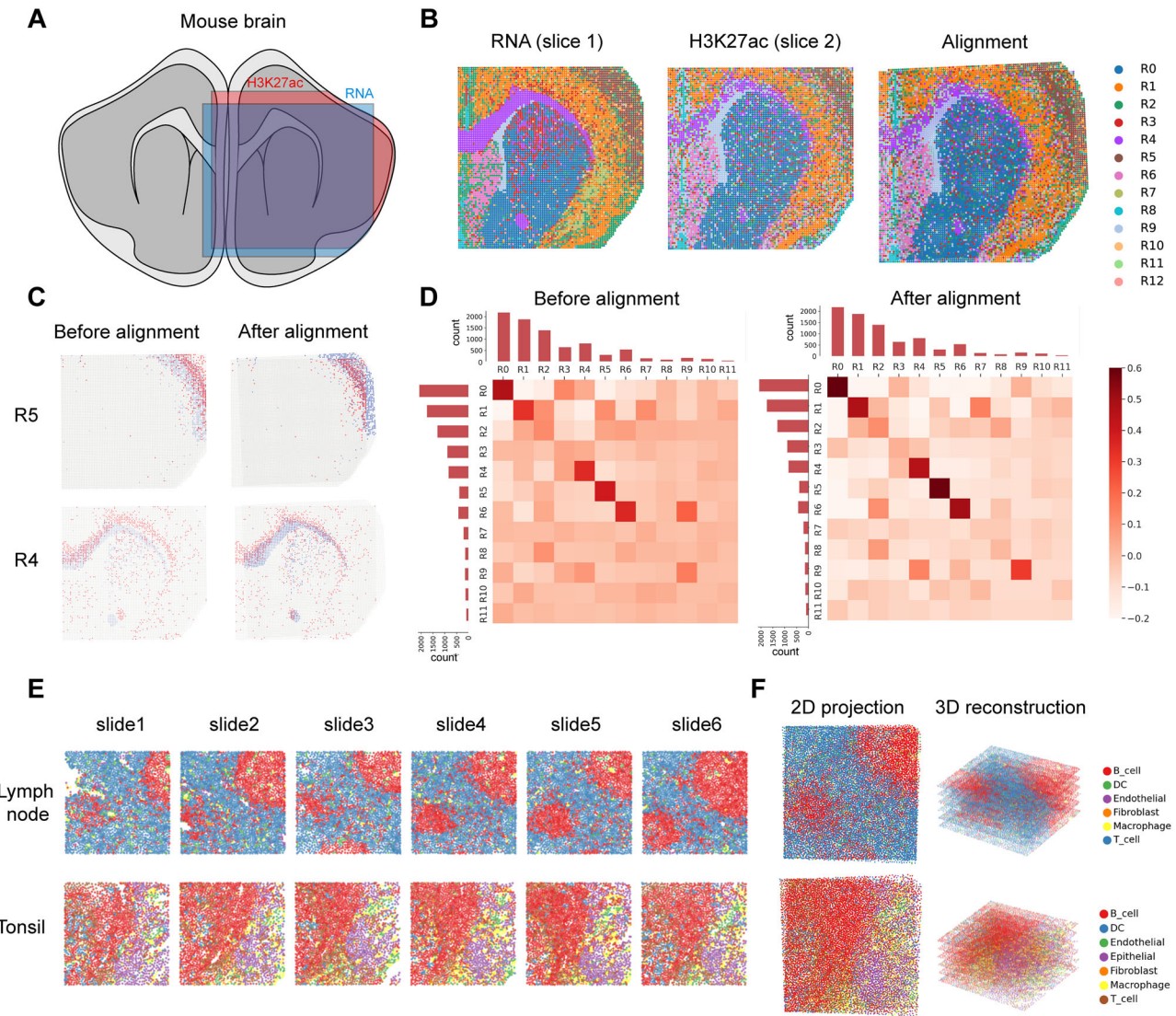

**Fig. 5 | Cross-modality alignment by SANTO. A** The spatial position of RNA and H3K27ac slices in the mouse brain. **B** The visualization of RNA and H3K27ac slices, and their alignment. **C** The visualization of distributions of two clusters before and after alignment. **D** Heatmaps of cluster correspondences before and after alignment. Bar plots on the axis represents the counts of all clusters. After alignment, the diagonals were distinctly darker, which indicates a better separation. **E** The visualization of 6 serial slices of MIBI-TOF datasets from human lymph node and tonsil. **F** 3D reconstructions and 2D projections of two MIBI-TOF datasets.

align two embryos with spatially reasonable positions. SLAT reconstructed less reasonable positions than ours with worse spatial translation. Regarding the results from PASTE and STAligner, they cannot reconstruct reasonable positions with impossible 3D spatial transformation. We also evaluated the performance of these results by PCC and CI, and our methods performed much better PCC and CI than other three methods (Supplementary Fig. 15).

**Cross-modality alignment on multiple spatial omics**

SANTO is generalizable to other spatial omics, enabling cross-modality alignment. Here, we start by illustrating the capability of SANTO in cross-modality alignment using two slices from the juvenile mouse brain including separate spatial transcriptomic and epigenomic data sequenced by Spatial CUT&Tag–RNA-seq (Fig. 5A)[31,32]. The epigenomics data measured the histone modifications called H3K27ac, which was associated with the higher activation of transcription and defined as an active enhancer marker[33]. We unify the common features of the two modalities by embedding gene expression from RNA-seq and gene score from H3K27ac into the same latent space, and align two slices together by SANTO (Fig. 5B).

For R4 and R5, they were clearly mismatched before alignment, and SANTO could reconstruct the precise alignment of them (Fig. 5C). By exploring the cell-type mapping between the two slices for each pair of adjacent spots, we find that our alignment improved the matching between cell types from two slices (Fig. 5D). We also try to identify the relationship between the clarity of anatomical structure and accuracy of alignment for each cell type. We firstly calculated the entropies of all cell types, which denoted the heterogeneity of each anatomical structure. Lower entropy represented a clearer anatomical structure. And the PCC between the clarity of anatomical structure and accuracy of alignment for each cell type is −0.69 (Supplementary Fig. 16, Supplementary Table 1). It concludes that the lower entropy (the clearer anatomical structure, e.g., R0, R1, R4, R5) is highly correlated with higher accuracy of alignment.

Lastly, SANTO is also applicable to align other omics data, even with few features. We use SANTO to align the spatial proteomics data from MIBI-TOF including six serial slices from lymph node and tonsil (Fig. 5E)[34]. Both datasets contain 16 phenotypic markers. SANTO successfully reconstructed both datasets as 3D-level profiles, revealing more comprehensive spatial patterns of phenotypes in the 3D level. By

projecting them into 2D, we observe that SANTO reveals consistent spatial patterns of cell types across the slices (Fig. 5F).

## Discussion

In this study, we propose SANTO to solve both alignment and stitching tasks for spatial omics as a general-purpose unified framework. SANTO uses a coarse-to-fine strategy to rapidly identify reasonable initial positions for two slices and refine their final alignment by their fine-grained features. Comprehensive benchmarking reveals superior performance of SANTO with respect to accuracy, robustness and usability. We further demonstrate the power of SANTO on various challenging applications, including cross-platform stitching, 3D-to-3D spatiotemporal alignment and cross-modality alignment.

SANTO stitches cross-platform slices of breast cancer to understand TME better with their complementary features. Stitching multiple partially overlapped slices by the same platform enables reconstruction of a comprehensive molecular profile with a larger view, which will be particularly useful on the research of large tissues, such as tissues from large mammalian and grown tumors in the TME. On the other hand, stitching of cross-platform slices can take advantage of different strengths of various spatial omics technologies, such as genomic coverages and spot resolution, and generate synergistical effect that reconstructs a panoramic view with genome-wide profiling at subcellular resolution. In our application, we stitch cross-platform slices from Xenium and Visium, enabling rich and precise characterizations of tumor ecosystems, which help us identify the immune cells undetected before (CD4 T cells, CD8 T cells and CD163 macrophages) for Visium data and enlarge the limited gene coverage of Xenium data. It fundamentally improves our knowledges of tumorigenesis and offers routes for therapeutic interventions[7,35].

To our knowledge, SANTO is the first method enabling 3D-to-3D spatiotemporal alignment to study tissue development. Spatio-temporal omics are crucial to understand the molecular dynamics and spatial dependencies in the biological process or system[36]. In our cases, we align two 3D-level mouse embryos from two timepoints to study the spatiotemporal transformation of tissues. Other than tissue development, spatiotemporal alignment by SANTO would also contribute to the studies of aging and cancer evolution. Aging is a predominant risk factor for cognitive dysfunction and many neurodegenerative disorders, whose relationship with these diseases is still unclear[37,38]. However, alignment of spatiotemporal omics across different timepoint would supply a quantitative understanding of dynamics in the aging and provide new insights of these relationships[37]. In the study of cancer etiopathogenesis, alignment of spatiotemporal omics possess the ability to characterize the molecular changes of biomarkers and decode spatiotemporal heterogeneity underlying tumor invasion, metastasis, antitumor response[39,40].

SANTO can also properly align cross-modality spatial omics data. Different omics technologies have revolutionized various aspects of molecular cell biology research[41]. However, studying complex living systems need the integration of different omics data with their complementary coverage[42]. In our application, we align the spatial transcriptomics and epigenomics data together from the mouse brain. Under the proper cross-modality alignment, the spatial gene regulatory work can be identified further with spatial transcriptomics and epigenomics data. Since more spatial omics platforms are emerging, cross-platform alignment of different omics with complementary features would create a holistic understanding of cells, organisms, and communities.

Moving forward, several aspects that warrant further exploration include: (1) beyond the rigid spatial transformation, non-rigid distortion of spatial domains can be further considered during alignment or stitching of slices with morphological difference. (2) through stitching task, limited overlap region is much less informative to stitch the slices well. Our experiments conclude that the stitching results by SANTO are less reasonable if percentages of overlap regions are below 20% (Supplementary Fig. 6). If possible, using corresponding pre-aligned histology images of two slices as additional feature can help stitch them well. (3) under precise stitching or alignment of spatial omics data, there is a wealth of directions for modeling the dynamics of space and time scale. For example, in the field of precision medicine, decoding the molecular dynamics of diseased tissues from different patients could potentiate the differential diagnosis, prognosis, personalized risk assessment and treatment[40,43]. And with proper modeling of spatiotemporal omics data, it is valuable to develop computational methods to forecast the future outcomes of molecular profiles, especially for treatment response and prognosis according to previous and current states of diseases. It will provide the prophetic in silico feedback for improving decision-making from physicians.

## Methods

### SANTO

In general, each spatial omics dataset contains $N$ cells/spots, and the spatial coordinates and omics feature expression profiles are denoted as $\mathscr{S} \in \mathbb{R}^{N \times D}$ and $\mathscr{G} \in \mathbb{R}^{N \times F}$, respectively, where $D$ represents the dimension of spatial coordinates that equals to 2 (x and y axis) or 3 (x, y and z axis) and $F$ denotes the number of omics features shared among all of cells/spots, e.g., number of genes in spatial transcriptomics data. In the alignment and stitching tasks, there are a source slice $X$ and a target slice $Y$. $X$ contains $\mathscr{S}_X$ and $\mathscr{G}_X$ with $N_X$ cells/spots and $F_X$ omics features in $\mathscr{G}_X$. Similarly, $Y$ contains $\mathscr{S}_Y$ and $\mathscr{G}_Y$ with $N_Y$ cells/spots and $F_Y$ omics features in $\mathscr{G}_Y$. The goal of SANTO is to align/stitch the spatial coordinates between slice $X$ and slice $Y$ properly.

**Coarse alignment.** Sinces $X$ and $Y$ have non-identical omics features, the common features $F$ is the intersection between $F_X$ and $F_Y$, then $\mathscr{G}_X$ and $\mathscr{G}_Y$ are extracted as the shape of $N_X \times F$ and $N_Y \times F$. Next, two omics feature expression profiles $\mathscr{G}_X$ and $\mathscr{G}_Y$ are used to calculate the Pearson Correlation Coefficient (PCC) between each pair of the $i$-th and the $j$-th cells/spots $X_i$ and $Y_j$ from slices $X$ and $Y$:

$$PCC\left(\mathscr{G}_{X_i}, \mathscr{G}_{Y_j}\right) = \frac{\sum \left(\mathscr{G}_{X_i} - \bar{\mathscr{G}}_X\right)\left(\mathscr{G}_{Y_j} - \bar{\mathscr{G}}_Y\right)}{\sqrt{\sum \left(\mathscr{G}_{X_i} - \bar{\mathscr{G}}_X\right)^2 \sum \left(\mathscr{G}_{Y_j} - \bar{\mathscr{G}}_Y\right)^2}}, \quad (1)$$

where $\bar{\mathscr{G}}_X$ and $\bar{\mathscr{G}}_Y$ represent the mean values of $\mathscr{G}_X$ and $\mathscr{G}_Y$ among omics features. With output PCC matrix with the shape of $N_X \times N_Y$ between $X$ and $Y$, each cell/spot $X_i$ can be paired with a specific cell/spot $Y_j$ if $PCC(\mathscr{G}_{X_i}, \mathscr{G}_{Y_j})$ has the highest PCC score among PCC scores between $X_i$ and all of cells/spots in $Y$. This strategy is under the consideration that most of pairs are spatially proximal, which can dominate the proper transformation even if few pairs are not spatially proximal.

Alternatively, for the stitching task, the overlap region between $X$ and $Y$ needs to be identified by filtering all pairs inside the overlap region. Thus, we firstly sort all PCC values of pairs with the descending order. To identify the pairs inside the overlap region, we formulate this problem as change point detection. We want to find a dramatic change in value as the change point (an index of pair), which can distinguish the most difference between left and right points. And we will keep the left points because the highly correlated pairs are more likely to be concentrated in the overlap region. For each pair along the indices, we calculate the cost for each pair $x$ by $cost(x_{left}) + cost(x_{right})$, where $x_{left}$ and $x_{right}$ represent all left and right pairs of the current pair $x$ along the indices. This cost quantifies the integrated variances of left and right pairs for each index $x$ along the indices. For the set of pairs $x_I$ (i.e., $x_{left}$ or $x_{right}$), $cost(x_I)$ is to quantify the variance of PCCs among a set

of pairs $x_I$, which is defined as:

$$cost(x_I) = \sum_{t \in I} ||p(x_t) - median(p(x_I))||_1, \quad (2)$$

where the $t$-th pair $x_t$ in $x_I$ has the PCC value as $p(x_t)$, and $median(p(x_I))$ is the median value of PCCs from all pairs in $x_I$. After calculating the cost of all pairs along the indices, we can finally find the change point $x'$ with the minimum cost, which is the best partition along the indices to distinguish that the pairs of cells/spots are inside or outside the overlap region. Next, we delete the right pairs and keep the left pairs of the best partition $x'$. Then, for each $i$-th cell/spot $X_i$ in $X$ and its paired cell/spot $Y_{i_{pair}}$ in $Y$, they can be used to calculate the covariance matrix $H$ between $X$ and $Y$ given by:

$$H = \sum \left( \mathscr{S}_{X_i} - \bar{\mathscr{S}}_X \right) \left( \mathscr{S}_{Y_{i_{pair}}} - \bar{\mathscr{S}}_Y \right), \quad (3)$$

where $\bar{\mathscr{S}}_X$ and $\bar{\mathscr{S}}_Y$ denote the spatial centroids of $X$ and $Y$, respectively. Next, singular value decomposition (SVD) is used to decompose $H = U\Sigma V^T$ where $U$ and $V$ are unitary matrices and $\Sigma$ is a diagonal matrix with non-negative real numbers on the diagonal. After the SVD factorization, we can optimize the proper rotation and translation in closed-form by:

$$R = VU^T, \quad (4)$$

$$T = -R\bar{\mathscr{S}}_X + \bar{\mathscr{S}}_Y. \quad (5)$$

Finally, the spatial coordinates of source slices $\mathscr{S}_X$ can be updated as $\mathscr{S}'_X = R\mathscr{S}_X + T$, which is the result of the coarse alignment.

**Fine alignment.** Firstly, for both $X$ and $Y$, we construct $k$-NN graphs for cells/spots in $\mathscr{S}$ and $\mathscr{G}$ separately by Euclidean distances of spatial coordinates and omics feature expressions, respectively, where $k$ is a hyperparameter. Then, we use dynamic graph CNN (DGCNN) to learn permutation-invariant embeddings of each cell's/spot's spatial feature $\mathscr{S}$ and omics feature $\mathscr{G}$ in $X$ and $Y$ by the graphs we construct[22]. There are two embedding networks via DGCNN focusing on spatial feature and omics feature ($\mathscr{S}$ and $\mathscr{G}$) separately with different weights. On the other hand, since embedding networks focusing on each feature type (spatial feature or omics feature) in $X$ and $Y$ try to learn the same knowledge of corresponding feature type, the parameters of embedding networks for each feature type between $X$ and $Y$ are shared[23]. The initial input for the embedding networks is each constructed graph for both $\mathscr{S}$ and $\mathscr{G}$ in $X$ and $Y$. Through the forward mechanism of the embedding network, $X_i^l$ represents the embedding of the $i$-th cell/spot $X_i$ in the $l$-th layer in the graph, with $L$ layers in total. $h_\theta^l$ is the convolution layer in the $l$-th layer with weight $\theta$, differing by the feature type (spatial or omics features). $f$ denotes the channel-wise aggregation function $max$ and the final embedding $X_i$ in $\mathscr{S}$ or $\mathscr{G}$ is:

$$X_i = \|_{l=1}^L f \left( \left\{ h_\theta^l \left( X_i^{l-1}, X_j^{l-1} \right), \forall j \in \mathscr{N}_i \right\} \right), \quad (6)$$

where $\|$ denotes the concatenation of outputs from $L$ layers, $\mathscr{N}_i$ denotes the neighbors of vertex $X_i$ in the graph and $j$ is the index of all of neighbors in $\mathscr{N}_i$. Through the forward pass each time, we recompute the input $k$-NN graph before the embedding network. This dynamic graph update learns how to construct the graph in each layer dynamically rather than a fixed graph through the training process.

After obtaining the separate embedding $\mathscr{F}_{\mathscr{S}}$ and $\mathscr{F}_{\mathscr{G}}$ of $\mathscr{S}$ and $\mathscr{G}$ for $X$ and $Y$, respectively, we have two final embeddings $\mathscr{F}_X$ and $\mathscr{F}_Y$ of $X$ and $Y$ that each embedding is the concatenation of embedded spatial features and omics features ($\mathscr{F}_{\mathscr{S}}$ and $\mathscr{F}_{\mathscr{G}}$) in the feature-wise from the

corresponding slice. Next, we generate a probabilistic mapping from $X$ to $Y$ by:

$$m(X,Y) = softmax \left( \mathscr{F}_Y \mathscr{F}_X^T \right). \quad (7)$$

The shape $m(X,Y)$ is $N_X \times N_Y$ and for each cell/spot in $X$ with index $i$, we have a soft pointer $m(X_i,Y)$ from $X_i$ into all cells/spots of $Y$ in $m(X,Y)$. With soft mapping between $X$ to $Y$, $\mathscr{S}'_X$ can be guided to transform to final coordinates $\mathscr{S}''_X$ by:

$$\mathscr{S}''_X = \mathscr{S}_Y^T m(X,Y), \quad (8)$$

where cells/spots with the same index in $\mathscr{S}'_X$ and $\mathscr{S}''_X$ are paired. Thus, we can optimize the final rotation and translation of the fine alignment from $\mathscr{S}'_X$ to $\mathscr{S}''_X$ by SVD as the previous demonstration (Eqs. 3, 4 and 5) and the spatial coordinates of the source slice $X$ can be finally aligned/stitched with the target slice $Y$ by rotations and translations from the coarse and fine alignment.

Since the fine alignment is an unsupervised method, we design the loss by co-considering spatial coordinates and omics feature expressions of cells/spots, under the assumption that proximal cells/spots between $X$ and $Y$ tend to be similar at spatial and omics feature level. We use a hyperparameter $\alpha$ to regulate the weight of loss between $\mathscr{S}$ and $\mathscr{G}$. The loss is defined as:

$$Loss = \frac{1}{N_X N_Y} \left( \sum_i^{N_X} softmin \cdot \sum_j^{N_Y} \left( \alpha \left( 1 - PCC \left( X_i, Y_j \right) \right) + (1-\alpha)Dis\left( X_i, Y_j \right) \right) \right) \quad (9)$$

where $PCC\left( X_i, Y_j \right)$ and $Dis(X_i, Y_j)$ denote the PCC values of omics feature expressions and Euclidean distance of spatial coordinates between the $i$-th cell/spot $X_i$ and the $j$-th cell/spot $Y_j$. The $softmin$ is:

$$softmin = \frac{e^{-Dis(X_i,Y_j)/\tau}}{\sum_j^{N_Y} e^{-Dis(X_i,Y)/\tau}}, \quad (10)$$

which obtains the attention weights between each $X_i$ to $Y_j$. $\tau$ represents the temperature, which is used to regulate sensitivity of the $softmin$ function. Through the definition of the $softmin$ function, spatially closer pairs of cells/spots between $X$ and $Y$ would have larger attention weights. $softmin$ is used to consider the mapping between the $i$-th cells/spots $X_i$ in $X$ to a region of cells/spots in $Y$ (one-to-many soft mapping) instead of one-to-one hard mapping.

### Benchmarking
**Measurements.** We use three metrics to benchmark SANTO with other methods: PCC, CI and AAS. For PCC, we calculate the Pearson correlation coefficient of each pair of cell/spot $X_i$ in $X$ with index $i$ and its spatially closet cell/spot $Y_j$ with index $j$ after alignment, and take the average through all cells/spots in $X$:

$$PCC = \frac{1}{N_X} \sum_{N_X}^i PCC\left( X_i, Y_j \right). \quad (11)$$

To evaluate the performance in the cell-type scale, we calculate CI between $X$ and $Y$ with their original cell-type annotation. For each pair of cell/spot $X_i$ in $X$ with index $i$ and its closet cell/spot $Y_j$ with index $j$, if they have the same cell type, $I(X_i,Y_j)$ is 1; otherwise, $I(X_i,Y_j)$ is 0. CI takes the average of $I$ through all cells/spots in $X$, which is:

$$CI = \frac{1}{N_X} \sum_{N_X}^i I\left( X_i, Y_j \right), \text{ and } I\left( X_i, Y_j \right) = \begin{cases} 1, C_{X_i} = C_{Y_j} \\ 0, C_{X_i} \neq C_{Y_j} \end{cases}, \quad (12)$$

where $C_{X_i}$ and $C_{Y_j}$ denote the cell type annotations of cells/spots $X_i$ and $Y_j$.

Since in the tasks with simulated data, we manually rotate the source slice, the performance of the predicted rotation could be evaluated based on the ground-truth rotation. AAS calculates the gap between predicted and ground-truth rotation as:

$$AAS = 360° - \left| Angle_{pred} - Angle_{gt} \right|, \qquad (13)$$

where a higher AAS corresponds to a better prediction.

**Compared methods.** We use PASTE (v1.4.0) for comparing the alignment results with its default hyperparameters ('alpha' = 0.1). In the comparison with PASTE2 under the stitching task, we use the default 'alpha' as 0.1 and set the 's' as 0.3. Since SLAT and STAligner focus on one-to-one alignment, we use output one-to-one matching between two slices to calculate rotation and translation by SVD optimization via Eqs. 3, 4 and 5. For SLAT, we use the default 'k_cutoff' as 10 in the function 'Cal_Spatial_Net'. For STAligner, we use the default 'knn_neigh' as 50 in the function 'train_STAligner'. The benchmarking is conducted on a workstation with a 2 Intel(R) Xeon(R) CPU E5-2680 v3 @ 2.50 GHz (30,720 KB cache size; 24 cores in total) and 528 GB of memory. The GPUs are two Nvidia Quadro M6000 24 GB (48 GB in total). The operating system used is Ubuntu 18.04.

**Robustness.** In the evaluation of robustness, we select 3 different values of hyperparameters for learning rate, epoch, k and alpha (learning rate = 0.01, 0.05 and 0.1, epoch = 50, 100 and 200, k = 5, 10 and 15, alpha = 0.1, 0.3 and 0.5).

**Downstream analysis for breast cancer**
By getting scFFPE-seq data as the reference, we deconvolve the Visium data from breast cancer by RCTD with its default parameters[27,44]. After alignment between Xenium and Visium slices, we also deconvolve the Visium data by the Xenium data. And for each spot in the Visium data, we take the spatial coordinates of each spot as the center and circle with the radius as 30 to include cells from the Xenium data. If no Xenium cells are inside the circle, we choose the 3-NN cells in the Xenium data. Then we can annotate the cell type of each Visium spot by the dominating cell type among the chosen cells from the Xenium data. To explore the cell-cell communication in the tumor microenvironment, we use COMMOT (v0.0.3) by choosing the reference dataset as CellChat and the species as human[28,45].

To predict the expression levels of the uncharacterized genes of the Xenium data outside the overlap region, we firstly pair each Xenium cell in the overlap area with its spatially closet Visium spot. Then, according to all of pairs between Xenium and Visium in the overlap region, we fit all gene expressions of Xenium cells with uncharacterized target gene's expressions of the paired Visium spots by Poisson regression (sklearn.linear_model.PoissonRegressor)[46]. After fitting the model, we input all gene expressions of Xenium cells outside the overlap region to predict the expressions of the uncharacterized gene.

**3D-to-3D spatiotemporal alignment and cross-modality alignment**
In the 3D-to-3D spatiotemporal alignment, we choose 8 mouse embryonic slices from two time points E14.5 and E15.5, and align 4 slices in the same time point firstly to generate a 3D-level point cloud of cells. Because the extreme large-scale memory consuming with these millions of cells during spatiotemporal alignment, we bin these slices by 5 × 5 down-sampling.

In the cross-modality alignment, the omics features from the two slices are gene expression profile and gene score matrix from transcriptome and epigenome, respectively. Thus, we use Harmony

(max_iterations = 10) to unify two omics feature expressions with their PCA results (n_comp = 50) to the same hidden space for alignment further.

**Reporting summary**
Further information on research design is available in the Nature Portfolio Reporting Summary linked to this article.

## Data availability
All relevant data supporting the key findings of this study are available within the article and its Supplementary Information files. The datasets for STARmap PLUS (https://singlecell.broadinstitute.org/single_cell/study/SCP1375)[26], Visium (https://zenodo.org/records/4730634)[25], MERFISH (https://datadryad.org/stash/dataset/doi:10.5061/dryad.8t8s248)[24], Xenium and Visium (https://www.ncbi.nlm.nih.gov/geo/query/acc.cgi?acc=GSE243280)[27], stereo-seq (https://db.cngb.org/stomics/mosta/download/)[11], Spatial CUT&Tag–RNA-seq (https://www.ncbi.nlm.nih.gov/geo/query/acc.cgi?acc=GSE165217)[31,47] and MIBI-TOF (https://zenodo.org/records/5945388)[34] are publicly available from their original publications. Source data are provided with this paper.

## Code availability
SANTO is publicly available as a Python package at https://github.com/leihouyeung/SANTO[48].

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

## Acknowledgements

This publication is based upon work supported by the King Abdullah University of Science and Technology (KAUST) Office of Research Administration (ORA) under Award No REI/1/5234-01-01, REI/1/5414-01-01, REI/1/5289-01-01, REI/1/5404-01-01, REI/1/5992-01-01, and URF/1/4663-01-01. [H.L., W.He, W.Ha., X.X., C.X., X.G.], and supported in part by NIH grants R01 GM134005 and P50 CA196530 [Y.L., H.Z.].

## Author contributions

X.G. and H.L. conceived and initiated this study. H.L. and Y.L. designed the methodology. H.L., Y.L., W.He., W.Ha. and C.X. conducted all the experiments. H.L. and Y.L. outputted the figure and tables. H.L. and Y.L. wrote the manuscript under supervision of X.G. and H.Z. X.X. and E.G. polished the writing of manuscript. All authors are involved in discussion and finalization of the manuscript.

## Competing interests

The authors declare no competing interests.
