## [Peer Review File · Nature Communications]

SANTO: a coarse-to-fine alignment and stitching method for spatial omicsReviewer #1 (Remarks to the Author):

In this study, the authors developed a two-step computational method called SANTO for alignment and stitching tasks for spatial omics data. SANTO is a coarse-to-fine method with two main steps: it identifies reasonable positions and overlap regions using the similarity of expression profiles, and then refines the positions using a dynamic graph CNN considering both omics and spatial patterns. SANTO demonstrates superior performance over existing methods on benchmark datasets. SANTO shows promise for novel cell type identification and prediction in various alignment and stitching tasks. Overall, the manuscript is an important contribution to the field of spatial omics data analysis.

However, I believe that the following issues should be addressed before accepting this manuscript.

Major Comments:

(1) The authors benchmarked the methods by generating rotated slices for alignment. I do not see the necessity of using rotated slices. Take STAligner as an example. STAligner aligns the slices in latent space using a MNN method, and the embedding of STAligner is invariant to the rotation of the slices. Therefore, STAligner achieved very similar and consistent performance at different rotation angles. The rotations should also not affect the alignment of SANTO and SLAT. PASTE is based on the fused-GW method, which is also rotation invariant. I suspect that the reason PASTE has a poor performance is that it gets stuck on a local point.

(2) It is not surprising that SANTO and PASTA have higher PCCs in Figure 2B, since they both use the PCC information during alignment. To better compare the methods, the authors may need to evaluate the embedded aligned space with the spatial domain identification tasks using the ARI index.

(3) The authors proposed a new task, the stitching of slices, as a generalization of the alignment task without technical displacement. It seems to me that the stitching task is a partial alignment task as proposed in PASTA2. It is suggested to the authors to compare STAligner and SLAT on the stitching tasks (e.g. Fig2C).

(4) Ablation study: It appears that the probabilistic mapping $m(X,Y)$ in equation (7) in the fine mapping step does not rely on the coarse mapping step. An ablation study is needed to justify that coarse-to-fine mapping is better than coarse/fine mapping alone.

(5) Robustness of change point detection in PCC. PCC is calculated on the gene space, which can be distorted by the batch effect and noise. The authors have shown a smooth curve of PCC in Fig. 1A, while in reality it can be a very rough curve. I am curious if SANTO is robust by selecting pairs within the overlap regions.

(6) The dynamic graph CNN part is not clear to me. How it can extract the local and global patterns of spatial and omic? Are there advantages of dynamic graph CNN over other GNN frameworks like the graph attention technique used in STAligner?

Minor comments:

(1) SVD factorization: What are the advantages of the SVD method used in this manuscript compared to other point-cloud based alignment algorithms (e.g., ICP in PASTA and STAligner)?

(2) The authors argue that PASTE and PASTE2 are highly time consuming in the Introduction. However, Fig. 2H shows that PASTE2 is much faster than STAligner and even reaches a speed comparable to SANTO. In lines 228-229, "SANTO is nearly ten times faster than STAligner and eight times faster than PASTE2", which is not consistent with the results shown in Fig2H.

(3) "Figure 6D" in Line 340, "Figure 6E" in Line 345 and "Figure 6F" in Linea 350 are typos. Should be "Figure 5".

(4) Should "SR + T" in line 478 be corrected to "RS + T"?

(5) Some figures are not well annotated and explained. For example, in Fig. 2A, how are the two slices annotated with different colors? What do the colors mean?

(6) To be consistent with the order in which they appear in the main text, it is suggested that "stitching and alignment" in the title be changed to "alignment and stitching".

Reviewer #2 (Remarks to the Author):

I've attached my comments as a PDF file. Please see the attachment called SANTO-review.pdf.

Reviewer #2 (Remarks on code availability):

I've run the code and had multiple issues:

- conda no longer supports python 3.7 for newer mac computers (M1), this has to be changed
- I could not install the package without changing the setup file, the package versions were off. Also, the authors have a requirements file that is contradictory to the setup.py file
- after manually installing the dependencies I could get the package to run. However, it crashes if you do not have a GPU enabled device.

Concerning:

- While the code is available, I could not find any notebooks or similar that allows me to reproduce the results presented in the paper. This is something I'd like to request from the authors.

Reviewer #2 Attachment on the following page

SANTO - Review

Summary

In this manuscript, the authors introduce SANTO, a new method for aligning and stitching spatial omics data through a coarse-to-fine approach. They address the challenge of integrating spatial omics slices from various platforms and modalities. The method first identifies reasonable spatial positions and overlap regions between slices before refining their alignment by considering both spatial and omics patterns. In their experiments, SANTO demonstrates superior performance over existing methods in various tasks, including cross-platform stitching, 3D-to-3D spatiotemporal alignment, and cross-modality alignment. The paper presents multiple experiments and applications with the intention to showcase SANTO's robustness, speed, and ability to enhance the understanding of complex biological systems.

I'm not a biologist so I will focus my commentary on the computational aspects of this paper in the hope that some of the other reviewers having more expertise in this area.

Comments

- Could the users provide more specific use-cases of when stitching and alignment has resulted in novel biological insights. These methods have become quite popular recently, and often generate pretty figures, but I eager to see an example when the stitching and alignment is pivotal for advancing our understanding of the environment. Perhaps the authors could elaborate on this in their introduction.
- From what I understand, this is a method mainly designed to align serial sections, or at least sections of highly similar tissue slices. I believe it's uncommon to have more serial sections than in the higher 10's for most technologies right now, and at the same time spatial datasets are quite limited in their size. The question I'd like to pose is if it's not as efficient to just manually align the sections using, any interactive suite (e.g., Napari via

Squidpy). There seems to be no non-linear transforms and all of the alignments are quite obvious for a human. This approach does not scale well but I given the cost of spatial data and the effort it takes to generate it I don't anticipate massive datasets being generated anytime soon. There's also a propagation of error in the sequential pairwise alignment employed here. Essentially, I'd like the authors to address why human alignment is not sufficient.

- I'm also interested in the propagation of error, this will probably not be apparent unless a high number of sections are to be aligned. This dataset <https://www.molecularatlas.org/st-js-viewer> consists of multiple carefully aligned sections of the mouse brain (ST data, predecessor of Visium). If the method works well and is somewhat robust to noise, it should be able to successfully align these sections even if they all were perturbed (e.g., by rotation similar to what the authors apply in Figure 2A).
- I find the Method's section somewhat subpar. While the authors describe their initial coarse alignment step somewhat well, there's a clear lack of information in the part about the fine alignment, this makes it hard for me to evaluate their method. What is the DGCNN architecture (number of layers, activation function, normalization layers?). Are they using the original architecture or do they make changes. What are the node features in respective G and S graph, and how are the edge weights determined?
- Could the authors explain how to chose the appropriate hyperparameters if I don't have access to a ground truth dataset to evaluate on? Or is the claim from Figure 2F that the method is robust to the choice of hyperparameters? If this is the case perhaps the authors could show this on more than one dataset?
- What is the minimal number of recommended overlapping features between the datasets? A lot of the high-res spatial omics methods have features in the magnitude of the hundreds but certain protein datasets (e.g., CODEX) only have 20-40 features, is this still sufficient. I'd be really interested in seeing an ablation study looking at how the number of features impacts performance (for both SANTO and the other methods).
- How does the method behave with symmetric and/or repetitive data, for example the mouse olfactory bulb in this paper:

<https://www.nature.com/articles/s41467-022-29439-6>. If there are multiple similar elements in the spatial data, will the method still successfully align the sections?

- The authors state: *"This strategy is under the consideration that most of pairs are spatially*

proximal, which can dominate the proper transformation even if few pairs are not spatially proximal" about their correlation-based pairing. Could they support this with some quantitative metrics? One experiment would be to look at the proportion of instances where the two cells that are closest in GEX space are also closest (or in the K-NN) in the spatial space; this could be done in the same section or two manually aligned ones.

- Could the authors clarify exactly how the coarse alignments are incorporated into the fine alignment, I'm assuming the transformed coordinates of the former are used in the latter, but this was not fully clear to me.
- I want to say that I do think the authors have done a good job in proving that their method outperforms the existing baselines, however, I'm in general not very impressed with their (the other baselines) performance; hence, why I still have questions for the authors.
- I'd like the authors to elaborate a bit on the failure modes of their method. I believe it's equally important for a user to know when to *not* use a method as when it's recommended to be used.

Summary statement

While I do think this is an interesting contribution that might spark some interest in the omics-community, I do have some questions and concerns about the method as well as the impact. If the authors could successfully address these comments, I do think the manuscript is fit for publication - but I can't approve it in its current state.

Reviewer #3 (Remarks to the Author):

The paper introduces SANTO, an alignment method for multiple spatial omics slices. It involves two steps for alignment: the first step employs Singular Value Decomposition (SVD) for coarse alignment, and the second step utilizes Dynamic Graph CNN (DGCNN) for fine alignment. SANTO has been applied to several data scenarios, including slices with different conditions, cross-platform slices, 3D-to-3D spatiotemporal slices, and cross-modality slices, and presents notable results. However, the results should be presented more robustly, and the method also needs comprehensive benchmarking. My major concerns are as follows.

1. Lack of novelty (In the method part). The authors proposed a registration method jointly constituted of coarse alignment and fine alignment. Thereinto, the coarse operation is built upon connections retrieval of spots with the highest expression similarity. Published methods PASTE and PASTE2 use the same idea except for formulating the retrieval problem using Optimal Transport, which aims to achieve higher connection accuracy using global optimization. By SANTO, exceeding is less likely to be achieved by simply calculating the correlation between spots on slices and then filtering the connections. The fine alignment part of SANTO adopts DGCNN which dynamically updates graphs across different neural network layers. As graph graph-based embedding method was already adopted in SLAT, there is a lack of innovation in this part of the method design by SANTO. The value of the dynamical characteristics of DGCNN compared to other graph-based auto-encoders was not discussed by the authors.
2. Unnecessary step design. In coarse alignment, filtering by deviation was included after correlation calculation. However, as PCC already takes consideration of deviation as its normalization term (Equation (1), Methods), what is the necessity/meaning of recalculating deviation and doing the filtering? This step seems to be unnecessary.
3. The authors implemented a loss function operating on the premise that spatially adjacent spots are likely to demonstrate notable similarities in terms of both their spatial and omics-related attributes. Yet, in instances where two proximate spots possess divergent biological functions, could it be possible that the applied loss function inadvertently injects extraneous noise into the analysis? It seems that a better design can be applied to avoid the introduction of noise.
4. Lack of practical meaning in aligning cross-platform slices. The authors address the value of SANTO in cross-platform registration and its downstream application including cell type composition and gene imputation analysis. With two slices sequenced on different platforms covering different projected regions, the gene imputation analysis aims to prove the power of SANTO's registration by expanding Xenium's profile on spots it didn't cover. In this study, the problem scenario assumed by the authors is not valid since it can be avoided by sequencing both regions by Xenium. The real challenge encountered in sequencing is the inadequacies of a single method, hence expanding the sequencing result of a single method is not the solution.
5. Insufficient benchmarking. Concerning the task of 3D-3D spatiotemporal alignment, although there are no methods that declare their applicability to spatiotemporal datasets for alignment purposes, methods designed for two-slice alignment can also be adapted for 3D alignment. Therefore, It is imperative to conduct a comprehensive benchmark on the 3D alignment process, utilizing the appropriate metrics to evaluate performance accurately. What's more, the authors also need to declare the advance compared to one 3D dependent alignment not only on the results but also the performance of alignment.
6. In the final part of the result, it is stated, "Notably, cell types with clearer anatomical structures and higher content levels tended to be aligned more accurately." The authors should include experiment results to clarify the relationship between the accuracy rate range and the clarity of the anatomical structure.

Minor concern:

1. The author states in the Introduction, "But current technologies can only achieve the capture area up to 200 mm², which hinders the investigation of larger and unabridged slices dissected from huge tissues of mammalian species or TME." However, according to the paper titled "Single-cell spatial transcriptome reveals cell-type organization in the macaque cortex," the capture area can reach at least 15 cm². Therefore, I suggest that the Introduction should be revised.

We are very grateful to the three reviewers for their thoughtful and thorough comments, which definitely helped us improve our paper greatly. We have revised the paper following all of their comments. Below please find the point-by-point response to all the reviewers' comments.

Reviewer #1 (Remarks to the Author):

In this study, the authors developed a two-step computational method called SANTO for alignment and stitching tasks for spatial omics data. SANTO is a coarse-to-fine method with two main steps: it identifies reasonable positions and overlap regions using the similarity of expression profiles, and then refines the positions using a dynamic graph CNN considering both omics and spatial patterns. SANTO demonstrates superior performance over existing methods on benchmark datasets. SANTO shows promise for novel cell type identification and prediction in various alignment and stitching tasks. Overall, the manuscript is an important contribution to the field of spatial omics data analysis.

However, I believe that the following issues should be addressed before accepting this manuscript.

First of all, thank you so much for your support on our paper. We have followed all of your suggestions and comments to revise our manuscript, and we believe that it has greatly improved the quality of our paper. Below are the detailed responses to each of the comments.

Major Comments:

(1) The authors benchmarked the methods by generating rotated slices for alignment. I do not see the necessity of using rotated slices. Take STAligner as an example. STAligner aligns the slices in latent space using a MNN method, and the embedding of STAligner is invariant to the rotation of the slices. Therefore, STAligner achieved very similar and consistent performance at different rotation angles. The rotations should also not affect the alignment of SANTO and SLAT. PASTE is based on the fused-GW method, which is also rotation invariant. I suspect that the reason PASTE has a poor performance is that it gets stuck on a local point.

Thank you for the comment. The main goal of SANTO is to properly align or stitch slices into a common coordinate framework (CCF). To benchmark SANTO and other methods comprehensively, we have simulated multiple scenarios including different rotations, percentages of overlap regions, noises of gene expression profiles and number of genes. Rotation is one of the factors we considered in our simulations and through the benchmarking we found the performance from all methods indeed varied among different rotations of simulated datasets, although, as you pointed out, that they should all be rotation invariant in theory. For example, from the performance evaluated by PCC, CI and AAS on the STARmap PLUS, DLPFC and MERFISH datasets below, we could see that PCC, CI and AAS of all methods were different among different rotations in these datasets. Regarding the poor performance of PASTE, we totally agree with your comments. From our benchmarking results, PASTE cannot perform well especially on image-based datasets. We believe that PASTE should consider unbalance optimal transport, which makes

more sense, because PASTE uses balanced optimal transport, which assumes that two slices have identical mass.

(2) It is not surprising that SANTO and PASTE have higher PCCs in Figure 2B, since they both use the PCC information during alignment. To better compare the methods, the authors may need to evaluate the embedded aligned space with the spatial domain identification tasks using the ARI

index.

Thank you for this excellent suggestion. We evaluated the aligned embedding space with spatial domain identification task by ARI score on the DLPFC datasets compared with STAligner, SLAT and PASTE. For each method, we outputted the visualization of spatial domain identification, and the UMAP of domain annotation and our clustering results based on the aligned embedding space, then we also calculated the ARI score for each method (the figure below). From the results, we could see that our method achieves relatively high ARI compared with the other three methods, which means the aligned embedding space of our method could cluster different spatial domains. Furthermore, we need to clarify that SLAT and STAligner aim to embed the spots/cells into a hidden space, which is expected to split different cell types well and output one-to-one alignment between two slices. But our method aims to embed the spots/cells by integrating spatial and omics features together, and output a global plane-to-plane alignment between the two slices. The optimization of loss and the goal our method achieves are radically different with SLAT and STAligner, which would result in different expectations of embedding spaces between our method and others. However, even spatial domain identification is not our main goal, through the comparison above, our method also has the ability to distinguish different spatial domains based on the aligned embedding space.

(3) The authors proposed a new task, the stitching of slices, as a generalization of the alignment task without technical displacement. It seems to me that the stitching task is a partial alignment task as proposed in PASTA2. It is suggested to the authors to compare STAligner and SLAT on the stitching tasks (e.g. Fig2C).

Thank you for the great suggestion. Following the comments, we evaluated the performance of STAligner and SLAT on the stitching task for two DLPFC datasets (151507_151508 and 151669_151670). Similar with the benchmarking of our method and PASTE2, we simulated two DLPFC datasets with three kinds of overlap regions (80%, 60% and 40%) and three kinds of rotations (30°, 60° and 90°). Then, we visualized all of benchmark results shown in the first figure below, where we could observe that SLAT performed worst in terms of performance. And we also quantified the PCC, CI and AAS for the performance of all the methods, where our method performed best in general across all of conditions evaluated by PCC, CI and AAS.

(4) Ablation study: It appears that the probabilistic mapping $m(X,Y)$ in equation (7) in the fine mapping step does not rely on the coarse mapping step. An ablation study is needed to justify that coarse-to-fine mapping is better than coarse/fine mapping alone.

Thank you for the valuable comment. We supplied an ablation study including several conditions of STARmap PLUS datasets. We selected the STARmap PLUS dataset from 8-month mouse brain

and generated 6 slices by rotating the slice from 45° to 270° . Then our method was tested under coarse alignment only, fine alignment only and integration of them. We visualized the results in the following figures in which red slices were transformed source slices and blue slices were target slices. From the visualizations below, coarse alignment supplies a reasonable mapping under large rotations, but it cannot align two slices precisely since it uses gene expression profiles from two slices only. On the other hand, fine alignment cannot align two slices well without the coarse alignment, which highly relies on reasonable initial mapping of two slices. Thus, by integrating coarse and fine alignment, our method could rapidly supply a reasonable initial alignment first, and consider the high-level information from spatial coordinates and gene expression profiles to align the slices precisely. From the violin plot below (the second figure), since coarse or fine alignment cannot align slices accurately, their PCC and CI are also lower than integration of coarse and fine alignment. Especially, since the coarse alignment only relies on the gene expression profiles, its PCC and CI are always the same through different rotations, which are shown as two lines in the violin plot respectively.

(5) Robustness of change point detection in PCC. PCC is calculated on the gene space, which can be distorted by the batch effect and noise. The authors have shown a smooth curve of PCC in Fig. 1A, while in reality it can be a very rough curve. I am curious if SANTO is robust by selecting pairs within the overlap regions.

Thanks for raising this important comment. We used the DLPFC datasets (151507_151508 and 151507_151508) to evaluate the robustness of our method under different kinds of noises. We added different gaussian noises ($\mu=0.1, \sigma=0.01$; $\mu=0.2, \sigma=0.02$; $\mu=0.3, \sigma=0.03$) to gene expression profile of each dataset with different overlap regions (80%, 60% and 40%), and calculated the PCC and CI for different conditions. Then, we calculated the standard deviations of these two metrics to see the changes among different kinds of noises. From the bar plots below, the values represent the standard deviations of PCC (blue) and CI (brown) among three kinds of noises and no noise under different overlap regions including 80%, 60% and 40% (thresholds: 0.1, 0.2 and 0.3). We can observe that the PCC is stable with different noise levels. The standard deviation of CI is fairly low and the highest value is just around 0.01, which hardly affect the performance of CI, because the average performance of CI is around 0.9. We also visualized the stitching results after coarse alignment under different noises and overlap regions. From the visualization, almost all of conditions do not affect the stitching after coarse alignment, which means the selection of pairs in the overlap region by change point detection is quite stable.

We can also explain why our selection of pairs is stable. Firstly, we used K-Nearest-Neighbor (KNN) to hierarchically smoothen the gene expression profile to select stable pairs of spots/cells. Intuitively, for each spot/cell in the gene expression profile, we selected its K nearest neighbors, and added the average of their gene expressions to current gene expression of the spot/cell. Through different hierarchies with different Ks, we smoothened the gene expression profiles by enlarging the receptive field of each spot/cell. Thus, the selection of pairs based on smooth gene expression profiles would be not so sensitive as original gene expression profiles. Secondly, we also supplied two options to reduce the batch effects between slices. Before the coarse alignment, PCA can be used on each dataset to generate a concentrated representation which would reduce the sensitivity among high-dimensional gene expression profile. Other than that, algorithms of batch effect removal can be also applied before coarse alignment. In our method, we used a well-known method 'Harmony' [1] to reduce the batch effect between two slices. These mentioned strategies would help our method select stable pairs from the two slices.

151507_151508

151669_151670

(6) The dynamic graph CNN part is not clear to me. How it can extract the local and global patterns of spatial and omic? Are there advantages of dynamic graph CNN over other GNN frameworks like the graph attention technique used in STAligner?

Thank you for the questions. Let us clarify the extraction of local and global patterns first. To extract the local and global patterns of spatial coordinates and omics features, we concatenated the features of nodes (spots/cells) of all the layers from initial layers to final layers through the dynamic graph CNN (DGCNN). From the embeddings in the layers through DGCNN, the layers closer to initial layer contain more local patterns and the layers closer to final layer contain more global patterns. The extraction of local and spatial patterns for spots/cells are not directly benefited from DGCNN, it is a general idea for graph-based neural network to concatenate all layers to integrate local and global patterns.

Then, we would like to clarify the advantages of DGCNN. SLAT and STAligner are both graph-based methods, but they use fixed graph constructed by KNN from spatial Euclidean distances of spots/cells. Our method uses DGCNN, which allows the graph to update dynamically after each layer in the deep learning network instead of traditionally fixed graphs, according to the difference between proximities in the graphs of embeddings and original input. It breaks the restriction of

fixed graph construction that only use spatial coordinates to calculate KNN utilized by SLAT and STAligner.

Minor comments:

(1) SVD factorization: What are the advantages of the SVD method used in this manuscript compared to other point-cloud based alignment algorithms (e.g., ICP in PASTA and STAligner)?

Thank you for the question. There are two reasons why we used SVD factorization instead of ICP: 1) SVD factorization can directly optimize the proper transformation whereas ICP needs to iterate multiple steps to converge. 2) ICP is under an assumption that two slices should be roughly aligned first because it is a local optimization algorithm, which is not a common scenario because two slices may be spatially unaligned.

(2) The authors argue that PASTE and PASTE2 are highly time consuming in the Introduction. However, Fig. 2H shows that PASTE2 is much faster than STAligner and even reaches a speed comparable to SANTO. In lines 228-229, "SANTO is nearly ten times faster than STAligner and eight times faster than PASTE2", which is not consistent with the results shown in Fig2H.

Thank you for pointing this out. Since we benchmarked SLAT and STAligner in the stitching task, we updated the figure of time consumption as below. To prevent confusion, we changed the label colors of these methods more distinctively. Through the figure, SANTO (50.3s) is nearly eight times faster than PASTE2 (449.2s) on the DLPFC dataset. Regarding all of datasets, SANTO consumes 39.8s on average, whereas STAligner consumes 462.4s on average, which is nearly ten times longer than SANTO. Therefore, the demonstration in the results is consistent with what the figure reveals.

(3) “Figure 6D” in Line 340, “Figure 6E” in Line 345 and “Figure 6F” in Line 350 are typos. Should be “Figure 5”.

Thank you for your careful reading. We have changed these in the manuscript accordingly.

(4) Should “SR + T” in line 478 be corrected to “RS + T”?

Thank you for your careful reading. We have changed these in the manuscript accordingly.

(5) Some figures are not well annotated and explained. For example, in Fig. 2A, how are the two slices annotated with different colors? What do the colors mean?

Thank you for pointing this out. Green slices are the source slices before alignment and blue slices are the target slices. Our goal is to align the source slices to the target slices. And the red slices are the transformed source slices outputted from different methods. We have changed the captions of Figure 2A.

(6) To be consistent with the order in which they appear in the main text, it is suggested that "stitching and alignment" in the title be changed to "alignment and stitching".

Thank you for the suggestion. We have changed the title according to your advice.

=====

Reviewer #2 (Remarks to the Author):

I've attached my comments as a PDF file. Please see the attachment called SANTO-review.pdf.

Questions from attached PDF of reviewer #2

Summary

In this manuscript, the authors introduce SANTO, a new method for aligning and stitching spatial omics data through a coarse-to-fine approach. They address the challenge of integrating spatial omics slices from various platforms and modalities. The method first identifies reasonable spatial positions and overlap regions between slices before refining their alignment by considering both spatial and omics patterns. In their experiments, SANTO demonstrates superior performance over existing methods in various tasks, including cross-platform stitching, 3D-to-3D spatiotemporal alignment, and cross-modality alignment. The paper presents multiple experiments and applications with the intention to showcase SANTO's robustness, speed, and ability to enhance the understanding of complex biological systems.

I'm not a biologist so I will focus my commentary on the computational aspects of this paper in the hope that some of the other reviewers having more expertise in this area.

Thank you very much for your support and the constructive comments. We have followed all your suggestions and comments to revise our manuscript, and we believe that they helped improve the quality of our paper greatly. Below are the detailed responses to each of your comments.

Comments

1. Could the users provide more specific use-cases of when stitching and alignment has resulted in novel biological insights. These methods have become quite popular recently, and often generate pretty figures, but I'm eager to see an example when the stitching and alignment is pivotal for advancing our understanding of the environment. Perhaps the authors could elaborate on this in their introduction.

Thank you for the valuable advice. Following the comment, we have elaborated the following statements into the introduction. Biological processes happen in a spatial context, and the 3D arrangement of cells in a tissue has a profound effect on their functions [2]. The major goal of alignment and stitching slices is to reconstruct multiple serial slices into 3D arrangement, which makes exploration of 3D molecular characteristics possible. 3D spatial omics is pivotal for multiple fields, including developmental biology and tumor environment, where 3D spatial structure is especially important to study.

We would like to provide an example that 3D spatial omics help biologists understand the development of *Drosophila* [3]. This study applied stereo-seq to late-stage embryos and all stages of larvae, generating multiple serial 2D slices for each sample across multiple timepoints (E14-16, E16-18, L1, L2 and L3). For each timepoint, serial slices from each sample were spatially aligned and reconstructed to a 3D point-cloud-based model. By using the 3D models from multiple timepoints, this study used these to detect the functional subregions in embryonic and larval midgut, analyzed spatial cell state changes during larval spermatogenesis and identified active transcription factor regulons. Since 2D spatial omics data could not supply molecular information of slices over and under current slice, which impedes the study of 3D cell-cell communications in spatial context, 3D anatomical structure identification and development of organogenesis. This example demonstrates the necessity of designing alignment and stitching methods to reconstruct the 3D molecular profiles.

2. From what I understand, this is a method mainly designed to align serial sections, or at least sections of highly similar tissue slices. I believe it's uncommon to have more serial sections than in the higher 10's for most technologies right now, and at the same time spatial datasets are quite limited in their size. The question I'd like to pose is if it's not as efficient to just manually align the sections using any interactive suite (e.g., Napari via Squidpy). There seems to be no non-linear transforms and all of the alignments are quite obvious for a human. This approach does not scale well but I given the cost of spatial data and the effort it takes to generate it I don't anticipate massive datasets being generated anytime soon. There's also a propagation of error in the sequential pairwise alignment employed here. Essentially, I'd like the authors to address why human alignment is not sufficient.

Thank you for raising this concern. There are several reasons why human alignment is not sufficient. 1) Human alignment highly relies on the subjective decision, but current spatial omics technologies enrich even sub-cellular resolution of spots, for example, the recently developed sub-cellular spatial omics technology, stereo-seq, by BGI. In this case, even tiny rotation or translation that human ignores would compromise the reliability of alignment. In contrast, automated methods like SANTO can give a proper and objective alignment or stitching result by considering both spatial- and omics-level comprehensively. 2) Through the rapid development of spatial omics technologies with lower costs, sampling a large number of slices would be a common case to study 3D molecular profiles. For example, a recent study of *Drosophila* claimed that it sampled 1,151 slices from 57 samples [4]. In these massive datasets, manually aligning slices is highly time-consuming and extremely difficult to reconstruct samples accurately. However, using automated tools to align massive number of slices can help biologists improve efficiency and reproducibility and ensure accuracy, which allows biologists focusing on more creative activities.

3. I'm also interested in the propagation of error, this will probably not be apparent unless a high number of sections are to be aligned. This dataset <https://www.molecularatlas.org/st-js-viewer> consists of multiple carefully aligned sections of the mouse brain (ST data, predecessor of Visium). If the method works well and is somewhat robust to noise, it should be able to successfully align these sections even if they all were perturbed (e.g., by rotation similar to what the authors apply in Figure 2A).

Thank you for the constructive comment. It is truly important to discuss the propagation of error when reconstructing serial slices. We used the dataset you suggested including 40 slices of mouse brain with 34,503 spots and these slices are aligned according to their corresponding H&E images (row: original) [5]. Following your suggestion, we randomly rotated each slice from 0° to 30° and we visualized the 2D projections of distribution of spots according to the different domain annotation and slice id (row: random rotation). We applied our method on each adjacent pair of these slices and we visualized the 2D projection of alignment results (row: alignment) and the 3D reconstruction of spatial domains for these 40 slices with two kinds of angles (the second figure). From the alignment results, we can observe that the 3D reconstruction of spatial domains is fairly similar with original alignment by H&E images, but the propagation error through the alignment of multiple adjacent slices truly existed. Because these 40 slices are not spatially adjacent slices in the mouse brain which have gap among them and the number of slices is really high. Thus, we totally agree with your advice that propagation error is important to be focused on, and we think a method that does not rely on aligning just adjacent slices but considering global information is highly needed in the future.

3D reconstruction

To further evaluate the performance of aligning multiple serial slices with noise, we also used MERFISH datasets including 12 slices with random rotations of each slice from 0° to 45° , which are shown as first subfigure ('Original') in the first figure below. We compared our method with PASTE, SLAT and STAligner, and we can observe that our method could reconstruct the most reasonable spatial profile than other methods. Additionally, we would like to quantitatively evaluate the performance with PCC and CI for each adjacent pair of slices shown as the second box plot. From the evaluation, our method achieved better PCC and CI than other methods. From the visualization and quantitative results above, our method is more stable than other methods under the perturbation of random rotations through multiple serial slices.

- I find the Method's section somewhat subpar. While the authors describe their initial coarse alignment step somewhat well, there's a clear lack of information in the part about the fine alignment, this makes it hard for me to evaluate their method. What is the DGCNN architecture (number of layers, activation function, normalization layers?). Are they using the original architecture or do they make changes. What are the node features in respective G and S graph, and how are the edge weights determined?

Thank you for pointing this important issue out. We have elaborated more details about fine alignment in the manuscript. Dynamic Graph Convolutional Neural Network (DGCNN) [6] is an extension of traditional Graph Convolutional Network (GCN), which could dynamically update the graph during learning process instead of fixed graph. We designed two DGCNNs with two graphs for extracting features from omics and spatial coordinates separately. For both DGCNN architectures, a graph feature extractor is firstly used to dynamically update the neighbors of all nodes and then four layers are used including the modules of graph convolution, batch normalization and ReLU activation function. Through concatenation of all outputs from these four layers, we learn the local and global feature together to enter the last layer including graph convolution, batch normalization and ReLU activation.

For each spot/cell in spatial omics, we have its spatial coordinates and omics features. Thus, the node of graph G is spot/cell and feature of node represents spot/cell's omics feature vector. Similarly, the node of graph S is also spot/cell and feature of node represents spot/cell's spatial coordinate vector. Both of graphs G and S are constructed based on their node features and the edge weights of them are either 1 or 0 (neighbor or not) calculated by KNN.

- Could the authors explain how to choose the appropriate hyperparameters if I don't have access to a ground truth dataset to evaluate on? Or is the claim from Figure 2F that the method is robust to the choice of hyperparameters? If this is the case perhaps the authors could show this on more than one dataset?

Thank you for the important questions. As demonstrated in the manuscript, our method is robust with respect to the choice of hyperparameters. To evaluate the robustness of our method, we selected three different values for each hyperparameter (learning rate, epoch, k and alpha). We tested these hyperparameters on all benchmarking datasets we used including two STARmap PLUS datasets (8 month and 13 month), two DLPFC datasets and two MERFISH datasets (animal ID=1 and 2). Through the following figures including quantifications of performance, we could observe that the performance is robust across the six datasets, four hyperparameters with three values and two metrics (PCC and CI).

We also calculated the standard deviations of PCC and CI for all hyperparameters from the results shown in the figure below. We could also observe the robustness of our method across different hyperparameters and datasets. The most extreme deviation was only around 4%, which would rarely affect the performance.

Although our method is robust through the evaluation, we also would like to supply several suggestions of hyperparameter choice through different datasets based on our experience: 1) If users try to stitch the slices, we recommend higher alpha (0.5-0.9) to focus more on the transcriptional loss. On the contrary, lower alpha could be chosen during alignment task. 2) For learning rate and epoch setting, we recommend between 0.01 to 0.001 and from 20 to 40 based on our experience. 3) The choice of k is based on the resolution of spots, and basically, we recommend the value of k between 10 to 30. Higher resolution of spots could have higher k and *vice versa*.

6. What is the minimal number of recommended overlapping features between the datasets? A lot of the high-res spatial omics methods have features in the magnitude of the hundreds but certain protein datasets (e.g., CODEX) only have 20-40 features, is this still sufficient. I'd be really interested in seeing an ablation study looking at how the number of features impacts performance (for both SANTO and the other methods).

Thank you for the excellent comment. We evaluated the performance of all methods with different gene numbers including all genes (135 genes), 110 genes and 90 genes on the MERFISH dataset. Two box plots evaluated by PCC and CI separately are shown below. From the quantitative performance, all methods are relatively stable across different numbers of genes. Thus, the minimal number of overlapped features depends on the complexity of dataset including the spatial heterogeneity of slices or the number of cell types, which means that there would not be an absolute value of minimal number of features. For example, the reconstruction of MIBI-TOF dataset in our paper includes 16 features per cell only. But they are all phenotypic markers, which are informative to distinguish the different cell types. Additionally, this dataset is not heterogeneous from cell type's spatial distribution and it only has 6 cell types. Thus, MIBI-TOF dataset could be also reconstructed

well by these limited features. In summary, if a dataset includes a number of features which could be used to split the difference among cell types in the feature-level embeddings, it will be sufficient.

- How does the method behave with symmetric and/or repetitive data, for example the mouse olfactory bulb in this paper: <https://www.nature.com/articles/s41467-022-29439-6>. If there are multiple similar elements in the spatial data, will the method still successfully align the sections?

Thank you for the questions. We tried our method on the suggested datasets. The datasets included the spatial transcriptomics data from mouse olfactory bulb profiled by Slide-seqV2 and stereo-seq,

and we visualized them by clustering the spots via ‘scanpy.tl.leiden’ [7] (two figures on the top) to show the identified spatial domains. Since these two datasets were profiled by different technologies based on two different mouse samples, they had batch effects and clear morphological distortions of spatial domains inevitably. After alignment by our method, these two slices were aligned reasonably well and most of spatial domains were under the same spatial locations (two figures on the bottom). Due to the different tissue morphologies between two slices, some detailed spatial domains were not perfectly aligned. Regarding these two slices from Slide-seqV2 and stereo-seq, it is a reasonable result that most of spatial domains from two slices were aligned. The pairs from most of aligned spatial domains dominated the spatial transformation between them and it would inevitably sacrifice the performance of alignment in the detailed local domains. It would be even serious under the alignment of slices from two different biological samples, because our method is designed to reconstruct slices from the same biological sample, which share the same tissue morphologies. In this case, it is also interesting to develop a method to consider the large morphological difference of slices in the future.

8. The authors state: “This strategy is under the consideration that most of pairs are spatially proximal, which can dominate the proper transformation even if few pairs are not spatially proximal” about their correlation-based pairing. Could they support this with some quantitative metrics? One experiment would be to look at the proportion of instances where the two cells that are closest in GEX space are also closest (or in the K-NN) in the spatial space; this could be done in the same section or two manually aligned ones.

Thank you for the excellent suggestion. We followed your suggestion to evaluate the reliability of the selected pairs from coarse alignment. We used two DLPFC datasets with different overlap regions and visualized the selected pairs from our method (‘Connections between selected pairs’). Black lines link each selected pair of spots/cells. We can observe that most of pairs are parallel and

consistent with spatial organizations of slices. Then, after alignment by selected pairs, we calculated the Euclidean distances between all selected pairs and outputted density plots of them for each kind of overlap regions. From all density plots, nearly 90% of pairs have the Euclidean distances lower than 40 and the side length of tissue slices is around 400, which means that the Euclidean distances of 90% of pairs are lower than 10% of side length of tissue slice and it is too tiny to affect the stability of global alignment of slices. Thus, we could see that the correlation-based pairing in the coarse alignment can select the pairs which are both spatially adjacent and close in the gene expression profiles. It could select stable pairs to supply a reasonable initial alignment for fine alignment further.

9. Could the authors clarify exactly how the coarse alignments are incorporated into the fine alignment, I'm assuming the transformed coordinates of the former are used in the latter, but this was not fully clear to me.

Thank you for the comment. Let us clarify more about the incorporation between coarse alignment and fine alignment. Originally, source and target slices may be **very unaligned** under the same common coordinate framework (CCF), and coarse alignment aims to rapidly supply a reasonable initial position for source slice according to target slice by using gene expression profiles only. With a more reasonable initial position, fine alignment would refine the positions of slices by considering both omics and spatial patterns locally and globally through dynamic graph updates. Thus, the key bridge between coarse alignment and fine alignment is the supply of reasonable initial position for the source slice.

To further prove the utility of coarse alignment, we provided an ablation study to show the performance of coarse alignment only, fine alignment only and integration of coarse and fine alignment. We selected the STARmap PLUS dataset from 8-month mouse brain and generated 6 slices by rotating the slice from 45° to 270°. Then we tested our method under coarse alignment only, fine alignment only and integration of them. We visualized the results as following figures in

which red slices were transformed source slices and blue slice were target slices. From the visualization below, we can see that coarse alignment only can supply a reasonable mapping under large rotations, but it cannot align two slices precisely since it uses gene expression profiles from two slices only. On the other hand, fine alignment could not align two slices well without the coarse alignment, which highly relies on reasonable initial alignment. Thus, by integrating coarse and fine alignment, our method could rapidly supply a reasonable initial alignment first, and then consider the high-level information from spatial coordinates and gene expression profiles to finely align slices precisely. From the violin plot below (the second figure), since coarse or fine alignment cannot align slices accurately, their PCC and CI are also lower than integration of coarse and fine alignment. Especially, the coarse alignment only relies on the gene expression profiles, thus its PCC and CI are always the same through different rotations, shown as two lines in PCC and CI respectively.

10. I want to say that I do think the authors have done a good job in proving that their method outperforms the existing baselines, however, I'm in general not very impressed with their (the other baselines) performance; hence, why I still have questions for the authors.

We truly appreciate your recognition of our efforts on the benchmarking with other methods and we acknowledge your concerns regarding the overall performance of other methods. Alignment and stitching are quite new tasks in the field of spatial omics. And we have selected all existing methods to benchmark, even though several methods cannot be directly applied to our task (e.g., SLAT and STAligner are designed for one-to-one alignment). Besides the selected compared methods, several methods focusing on alignment cannot be selected for benchmarking, e.g., STalign aims to align slices together, but it requires the manual landmarks of two slices from users for affine alignment and focuses more on the local distortion through alignment, which is impossible to benchmark with our method. Their tasks are radically different. Finally, we are also grateful about the valuable comments from reviewers, which truly improve the quality of our paper.

11. I'd like the authors to elaborate a bit on the failure modes of their method. I believe it's equally important for a user to know when to not use a method as when it's recommended to be used.

Thank you for this excellent suggestion. There would be two kinds of failure cases for our method, which are also described in the Discussion section:

1) As question 7 proposed, our method did not consider the scenario that two slices have clearly morphological difference or distortion of spatial domains. Because our method is designed to reconstruct slices from the same biological sample with the same tissue morphologies. In this

case, the alignment would be dominated by most of spatial domains to generate a globally reasonable reconstruction and ignore the local distortion, which is especially significant for slices from different biological samples.

- 2) The other scenario our method may be failure is that the overlap region between two slices is too small for stitching methods. It is reasonable that lower overlap region supplies less information between slices to stitch them well. To support this statement, we visualized and evaluated the performance of stitching task with different overlap regions (10%-80% overlap) on two DLFPD datasets. From visualization and evaluation by PCC and CI, our method has great performance of most percentages of overlap regions (30%-80%), but through the further reductions of overlap percentages (10%-20%), the PCC and CI drop, and the visualization shows less reasonable stitching results than previous overlap regions. We can imagine that when the overlap percentage drops as 20% or lower, limited overlap region is much less informative to stitch and the identified spot pairs would become limited and sensitive. One possible solution is to use corresponding pre-aligned histology images of two slices to help stitch them well.

Remarks on code availability from Reviewer 2:

I've run the code and had multiple issues:

- conda no longer supports python 3.7 for newer mac computers (M1), this has to be changed

Thank you for catching this. We have changed the version of Python as 3.10 now.

- I could not install the package without changing the setup file, the package versions were off. Also, the authors have a requirements file that is contradictory to the setup.py file

Sorry for this inconvenience. We have checked the contradictory and updated our packages. Please follow the updated README in the github to deploy our method.

- after manually installing the dependencies I could get the package to run. However, it crashes if you do not have a GPU enabled device.

We have changed our code to be suitable for users who have CPU only. From the start of our method, our code will check the availability of GPU, if not, CPU would be used then. But we highly recommend that the users have their GPU to run our model. The CPU users would consume inevitably long time to run our model.

Concerning:

- While the code is available, I could not find any notebooks or similar that allows me to reproduce the results presented in the paper. This is something I'd like to request from the authors.

Thank you for your concern. We have uploaded the reproduced codes as multiple notebooks including the benchmarking of our methods with PASTE, PASTE2, SLAT and STAligner on the STARmap PLUS, DLPFC and MERFISH datasets.

Summary statement

While I do think this is an interesting contribution that might spark some interest in the omics-community, I do have some questions and concerns about the method as well as the impact. If the authors could successfully address these comments, I do think the manuscript is fit for publication - but I can't approve it in its current state.

Thank you so much for your acknowledgment and valuable comments, which truly improved our paper. We have carefully revised our paper following all of your comments and suggestions. We believe that the revision now has much better quality than the original submission.

Reviewer #3 (Remarks to the Author):

The paper introduces SANTO, an alignment method for multiple spatial omics slices. It involves two steps for alignment: the first step employs Singular Value Decomposition (SVD) for coarse alignment, and the second step utilizes Dynamic Graph CNN (DGCNN) for fine alignment. SANTO has been applied to several data scenarios, including slices with different conditions, cross-platform slices, 3D-to-3D spatiotemporal slices, and cross-modality slices, and presents notable results. However, the results should be presented more robustly, and the method also needs comprehensive benchmarking. My major concerns are as follows.

Thank you very much for your very helpful comments. We have followed all your suggestions and comments to revise our manuscript, and we believe that they have greatly improved the quality of our paper. Below are the detailed responses to each of your comments.

1. Lack of novelty (In the method part). The authors proposed a registration method jointly constituted of coarse alignment and fine alignment. Thereinto, the coarse operation is built upon connections retrieval of spots with the highest expression similarity. Published methods PASTE and PASTE2 use the same idea except for formulating the retrieval problem using Optimal Transport, which aims to achieve higher connection accuracy using global optimization. By SANTO, exceeding is less likely to be achieved by simply calculating the correlation between spots on slices and then filtering the connections. The fine alignment part of SANTO adopts DGCNN which dynamically updates graphs across different neural network layers. As graph graph-based embedding method was already adopted in SLAT, there is a lack of innovation in this part of the method design by SANTO. The value of the dynamical characteristics of DGCNN compared to other graph-based auto-encoders was not discussed by the authors.

Thank you for your comment on the novelty of our method. Please see our detailed accounts of the novelty and better performance of SANTO than the existing methods.

PASTE and PASTE2 are based on Optimal Transport (OT), which uses original gene expression profiles and spatial coordinates to optimize a proper transportation matrix between two slices. Despite PASTE and PASTE2's novelty of firstly applying OT to the spatial omics alignment problem, these methods do not consider the high-level information through gene expression and spatial coordinates, which results in the local optima. Moreover, PASTE assumes that two slices have identical mass (balanced OT), but the fact is that such assumption does not hold in most cases, which means that it should penalize the mass variation in the loss function (unbalanced OT). On the other hand, SANTO used DGCNN to dynamically embed features of gene expression and spatial coordinates separately, which learns the latent knowledge from them. Then SANTO co-considers these two sources of latent knowledge to optimize the transformation of slices by global and local pattern of their graphs. Through the following three figures including our benchmarking results for six datasets, we could clearly see the superior performance of SANTO over PASTE and PASTE2 for both accuracy and time-consumption.

On the other hand, SLAT and STAligner are graph-based methods, but they both use fixed graph constructed by K-nearest neighbors from spatial Euclidean distances. SANTO uses DGCNN, which allows the graph to update dynamically after each layer in the deep learning network instead of traditionally fixed graphs, according to the difference between proximities in the graphs of embedding and original input. It breaks the restriction of graph construction that only uses spatial coordinates to calculate K-nearest neighbors utilized by SLAT and STAligner. Furthermore, SANTO focuses on aligning or stitching slices into a common coordinate framework (CCF) but SLAT and STAligner focus on one-to-one alignment between two slices, which could be also applied on our task. Thus, we used SVD factorization to optimize the transformation of slices through their output spot pairs. Through the following three figures including our benchmarking results over 6 datasets, the performance of SANTO was superior over SLAT and STAligner under the evaluation of accuracy and time-consumption.

MERFISH

2. Unnecessary step design. In coarse alignment, filtering by deviation was included after correlation calculation. However, as PCC already takes consideration of deviation as its normalization term (Equation (1), Methods), what is the necessity/meaning of recalculating deviation and doing the filtering? This step seems to be unnecessary.

Thank you for raising this concern. We apologize for the unclear description in the original submission. Please allow us to clarify this. Regarding the first calculation of PCC for extracting the pairs of spots/cells, we considered deviation of the Euclidean distance from the gene expression profiles from two slices as you pointed out. Then, we sorted all of PCCs calculated above in a descending order and used these sorted PCCs to conduct change point detection. In the calculation of change point detection, we considered the deviation of the Manhattan distance from sorted PCCs to filter the pairs which should exist in the overlap region of two slices. These two deviations are thus completely different.

3. The authors implemented a loss function operating on the premise that spatially adjacent spots are likely to demonstrate notable similarities in terms of both their spatial and omics-related attributes. Yet, in instances where two proximate spots possess divergent biological functions, could it be possible that the applied loss function inadvertently injects extraneous noise into the analysis? It seems that a better design can be applied to avoid the introduction of noise.

Thank you for the excellent question. We have used softmin function to softly handle this problem. The softmin function aims to assign more possible paired spots/cells from two slices higher weights through the calculation of loss and paired spots with divergent biological functions tend to be assigned lower weights. This would enable soft mapping between spots/cells instead of one-to-one hard mapping that prevents the negative impact from pairs with divergent biological functions. Moreover, the loss function considers the PCC of gene expression profiles between spots/cells from

two slices and we assumed that higher PCC indicates more similar biological functions or cell types. If the pairs with divergent biological functions, they would have lower PCCs, which would be automatically penalized in the loss calculation. Additionally, since our goal is to align two slices globally into a CCF, proximate pairs with similar biological functions, which represent the majority of all the paired spots, would dominate the optimization of spatial transformation for two slices.

4. Lack of practical meaning in aligning cross-platform slices. The authors address the value of SANTO in cross-platform registration and its downstream application including cell type composition and gene imputation analysis. With two slices sequenced on different platforms covering different projected regions, the gene imputation analysis aims to prove the power of SANTO's registration by expanding Xenium's profile on spots it didn't cover. In this study, the problem scenario assumed by the authors is not valid since it can be avoided by sequencing both regions by Xenium. The real challenge encountered in sequencing is the inadequacies of a single method, hence expanding the sequencing result of a single method is not the solution.

Thank you for the comment. We need to clarify that the downstream applications including unseen cell type identification and gene imputation aim to prove the reliability of our alignment and stitching results among two Xenium slices and Visium slice. The datasets we obtained have the areas that do not intersect (as shown in the following figure) and due to the current limitations of Visium and Xenium technologies, this situation allows us to impute undetected genes for Xenium slices based on the good alignment and stitching among these slices. We believe that due to the limited capture area of Visium technology, these datasets did not include other additional Visium slices outside the region of the overlap area. We need to emphasize that we just considered this scenario as a way to prove the reliability of alignment and stitching among Xenium and Visium slices, which prove the accuracy of SANTO further. At the same time, we fully agree with you that we also look forward to a new spatial omics technology, whose spot could detect genome-wide genes with single-cell resolution and large capture area. That would advance the entire field of spatial omics.

Capture areas from H&E image

5. Insufficient benchmarking. Concerning the task of 3D-3D spatiotemporal alignment, although there are no methods that declare their applicability to spatiotemporal datasets for alignment purposes, methods designed for two-slice alignment can also be adapted for 3D alignment. Therefore, it is imperative to conduct a comprehensive benchmark on the 3D alignment process, utilizing the appropriate metrics to evaluate performance accurately. What's more, the authors also need to declare the advance compared to one 3D dependent alignment not only on the results but also the performance of alignment.

Thank you for the comment. Following your comment, we have compared our method with PASTE, SLAT and STAligner on the 3D-to-3D spatiotemporal alignment of mouse embryos. Two mouse embryos from two timepoints (E14.5 and E15.5) were used including four slices through z axis from each of embryo. We firstly used each method to align the four slices from E14.5 and E15.5 separately to get two 3D profiles of two timepoints. Then, we modified the PASTE, SLAT and STAligner so that they could align datasets of 3D spatial omics instead of 2D spatial omics. Basically, we let these methods output rotation and translation matrices in the 3D level based on the SVD factorization. Finally, we aligned two embryos in the 3D level and outputted the visualization of E14.5 (blue) and E15.5 (red) after alignment. From the visualization results below, we can observe that our method could perfectly align two embryos with spatially reasonable positions. SLAT reconstructed less reasonable positions than ours with worse spatial translation. Regarding the results from PASTE and STAligner, they cannot reconstruct reasonable positions with almost unreasonable 3D spatial transformation. We also evaluated the quantitative performance of these results by PCC and CI (the second bar plot), and our methods performed much better PCC and CI than other three methods.

For the reason of the poor performance of other methods, we believe that the nature of one-to-one alignment from SLAT and STAligner makes them sensitive to optimize an appropriate 3D spatial rotation and translation. Regarding PASTE, it does not consider the factor of global matching during alignment and is easy to get stuck in the local minima, which is more extreme in the 3D-to-3D alignment. On the other hand, the component DGCNN of our method is originally adapted from point-cloud alignment which is naturally a 3D-to-3D alignment problem. Additionally, we considered the performance of global alignment in the loss design. We believe that these are the main reasons for the superior performance of our method.

6. In the final part of the result, it is stated, "Notably, cell types with clearer anatomical structures and higher content levels tended to be aligned more accurately." The authors should include experiment results to clarify the relationship between the accuracy rate range and the clarity of the anatomical structure.

Thank you for the comment. Following your suggestion, we explored the relationship between clarity of anatomical structure and the accuracy rate of alignment. To quantify the clarity of anatomical structure, we firstly calculated the entropies of all cell types, which denoted the heterogeneity of each anatomical structure. Lower entropy represents a clearer anatomical structure. For each spot/cell in the alignment results, we used 10-NN to construct a graph and calculated the Shannon entropy of this graph based on the neighbor nodes' cell-type annotation. Thus, we had entropy for each spot/cell and we averaged the entropies according to all cell types, which were shown as the last table with the accuracy of each cell type. We also drew a scatter plot to visualize

the relationship between entropy and alignment accuracy of each cell type as the second figure. In the figure, we calculated the Pearson correlation between the entropy and alignment accuracy, and visualized the 95% confidence interval along with the regression. The Pearson correlation between alignment accuracy and entropy for all cell types is -0.69 , where the lower entropy (the clearer anatomical structure) is highly correlated with higher accuracy of alignment. Especially, the top-4 clear structures (R4, R0, R9 and R5 from the first figure) have corresponding high accuracies of alignment (the second figure). Thus, these mentioned results indicate the high correlation between clarity of the anatomical structure and accuracy of alignment.

Cell Type	Entropy	Accuracy
R4	0.781	0.53
R0	0.908	0.59
R9	1.176	0.37
R5	1.257	0.57
R8	1.307	0.07
R6	1.340	0.48
R10	1.359	0.02
R1	1.363	0.38
R11	1.370	0.02
R2	1.373	0.25
R3	1.409	0.06
R7	1.521	0.01

Minor concern:

1. The author states in the Introduction, "But current technologies can only achieve the capture area up to 200 mm², which hinders the investigation of larger and unabridged slices dissected from huge tissues of mammalian species or TME." However, according to the paper titled "Single-cell spatial transcriptome reveals cell-type organization in the macaque cortex," the capture area can reach at least 15 cm². Therefore, I suggest that the Introduction should be revised.

Thank you for the suggestion. We have revised this part in the introduction accordingly.

References:

- [1] I. Korsunsky *et al.*, "Fast, sensitive and accurate integration of single-cell data with Harmony," *Nat. Methods*, vol. 16, no. 12, pp. 1289–1296, 2019, doi: 10.1038/s41592-019-0619-0.
- [2] D. Bressan, G. Battistoni, and G. J. Hannon, "The dawn of spatial omics," *Science*, vol. 381, no. 6657, p. eabq4964, 2023, doi: 10.1126/science.abq4964.
- [3] M. Wang *et al.*, "High-resolution 3D spatiotemporal transcriptomic maps of developing *Drosophila* embryos and larvae," *Dev. Cell*, vol. 57, no. 10, pp. 1271-1283.e4, 2022, doi: 10.1016/j.devcel.2022.04.006.
- [4] M. Wang *et al.*, "A single-cell 3D spatiotemporal multi-omics atlas from *Drosophila* embryogenesis to metamorphosis," *bioRxiv*, p. 2024.02.06.577903, Jan. 2024, doi: 10.1101/2024.02.06.577903.
- [5] O. Cantin, N. J. Fernandez, J. Aleksandra, M. Antje, L. Joakim, and M. Konstantinos, "Molecular atlas of the adult mouse brain," *Sci. Adv.*, vol. 6, no. 26, p. eabb3446, Sep. 2021, doi: 10.1126/sciadv.abb3446.
- [6] Y. Wang, Y. Sun, Z. Liu, S. E. Sarma, M. M. Bronstein, and J. M. Solomon, "Dynamic graph Cnn for learning on point clouds," *ACM Trans. Graph.*, vol. 38, no. 5, 2019, doi: 10.1145/3326362.
- [7] F. A. Wolf, P. Angerer, and F. J. Theis, "SCANPY: large-scale single-cell gene expression data

analysis," *Genome Biol.*, vol. 19, no. 1, p. 15, 2018, doi: 10.1186/s13059-017-1382-0.

Reviewer #1 (Remarks to the Author):

All my concerns have been addressed.

Reviewer #2 (Remarks to the Author):

I'm quite impressed by the diligence by which the authors answered my comments. They conducted multiple new experiments, which was a quite big time commitment, and followed all of my suggestions - showing me all the results that I was asking for.

The updated manuscript is of a much higher quality than the initial version and has a clearer message with more context.

I'm not yet fully convinced that these computational methods are competitive with human alignment, especially given the prevalence of error propagation. However, science is not always about perfection but also taking incremental steps towards it - and I do believe this manuscript constitute a valuable contribution for that sake.

I would recommend the revised version for publication.

Reviewer #2 (Remarks on code availability):

The authors have updated their repo and added the necessary notebooks for reproducibility. This is now in a production ready state.

Reviewer #3 (Remarks to the Author):

Thanks for the effort and revisions made by the author in the first round of review. I have carefully read the response letter and there are several my comments that still require further clarification and improvement. Specifically,

1. For my first comment, SANTO characterizes its coarse-to-fine registration framework, as well as its specially designed graph-based embedding learning towards fine alignment. Although the graph construction part of SANTO differs from existing methods, the authors are still required to demonstrate the effectiveness of these designs through practical scenarios. To address this concern, I suggest that the authors should conduct an ablation study on both the coarse alignment and the DGCNN module. Specifically, the coarse alignment procedure should be ablated to ascertain any differences in accuracy compared to the original SANTO. Furthermore, the fine alignment should be replaced by another existing method, such as SLAT, and then integrated with the originally adopted SVD, to measure any changes in alignment accuracy.

2. I would like to elaborate on the fifth comment. The authors should compare the registration results of E14.5 and E15.5 mouse embryos separately. Additionally, it would be beneficial to present the accuracy of SANTO in 3D data registration and compare it with other existing software to demonstrate its effectiveness in 3D registration. Furthermore, I noticed that there was no mention of PASTE2 in the author's response. Could you please explain this omission?

3. Regarding the sixth comment, it can be observed from the displayed accuracy that the majority of the mapping rates are less than 0.5. The authors may need to explain the reasons for the low accuracy or further demonstrate the feasibility of SANTO in cross-platform data mapping.

We are very grateful to the three reviewers for their positive comments on our revision. Below please find the point-by-point response to all the reviewers' comments.

Reviewer #1 (Remarks to the Author):

All my concerns have been addressed.

I am appreciated that you felt satisfied with our revision. Thanks for your comments on our manuscript.

Reviewer #2 (Remarks to the Author):

I'm quite impressed by the diligence by which the authors answered my comments. They conducted multiple new experiments, which was a quite big time commitment, and followed all of my suggestions - showing me all the results that I was asking for.

The updated manuscript is of a much higher quality than the initial version and has a clearer message with more context.

I'm not yet fully convinced that these computational methods are competitive with human alignment, especially given the prevalence of error propagation. However, science is not always about perfection but also taking incremental steps towards it - and I do believe this manuscript constitute a valuable contribution for that sake.

I would recommend the revised version for publication.

Thanks for your helpful comments which improved our study a lot.

Reviewer #2 (Remarks on code availability):

The authors have updated their repo and added the necessary notebooks for reproducibility. This is now in a production ready state.

Thanks for your acknowledgement of our code.

Reviewer #3 (Remarks to the Author):

Thanks for the effort and revisions made by the author in the first round of review. I have carefully read the response letter and there are several my comments that still require further clarification and improvement.

Thank you very much for your helpful and detailed comments. We have followed your comments to revise our manuscript and we hope the following response could fulfil your concern.

Specifically,

1. For my first comment, SANTO characterizes its coarse-to-fine registration framework, as well as its specially designed graph-based embedding learning towards fine alignment. Although the graph construction part of SANTO differs from existing methods, the authors are still required to demonstrate the effectiveness of these designs through practical scenarios. To address this concern, I suggest that the authors should conduct an ablation study on both the coarse alignment and the DGCNN module. Specifically, the coarse alignment procedure should be ablated to ascertain any differences in accuracy compared to the original SANTO. Furthermore, the fine alignment should be replaced by another existing method, such as SLAT, and then integrated with the originally adopted SVD, to measure any changes in alignment accuracy.

Thanks so much for your comments and specific suggestions. We have followed your suggestions to conduct the ablation study. We selected the STARmap PLUS dataset from 8-month mouse brain and generated 6 slices by rotating the slice from 45° to 270° . Then our method was tested under three conditions: no coarse alignment, replacing fine alignment with SLAT and our original method. We visualized the results in the following figures in which red slices were transformed source slices and blue slices were target slices. From the visualizations below, no coarse alignment cannot align two slices well without the coarse alignment, which highly relies on reasonable initial mapping of two slices. On the other hand, replacing fine alignment with SLAT has minor misalignment compared with our original method. Thus, by integrating coarse and fine alignment, our method could rapidly supply a reasonable initial alignment first, and consider the high-level information from spatial coordinates and gene expression profiles to align the slices precisely. From the violin plot below (the second figure), it is evident that since neither the 'no coarse alignment' nor the 'replacing fine alignment with SLAT' conditions can align slices accurately, their PCC and CI values are lower than those of our original method.

2. I would like to elaborate on the fifth comment. The authors should compare the registration results of E14.5 and E15.5 mouse embryos separately. Additionally, it would be beneficial to present the accuracy of SANTO in 3D data registration and compare it with other existing software to demonstrate its effectiveness in 3D registration. Furthermore, I noticed that there was no mention of PASTE2 in the author's response. Could you please explain this omission?

Thanks for your additional advice on 3D-to-3D alignment benchmarking. Following your suggestions, we have evaluated the performance during 3D registration of two timepoints separately. The first two bar plots include the benchmarking results of all methods evaluated by PCC and CI under two timepoints E14.5 (Blue) and E15.5 (Yellow). From the distributions of PCC and CI, we can observe that SANTO has fairly better performance than other four methods. We also recorded the consumed time by all methods during the entire alignment process, and found that SANTO and SLAT were highly effective. Following your advice, we also updated the results to include the performance of PASTE2. Through the updated visualization of 3D-to-3D alignment, we can see that PASTE2 cannot align 2D slices successfully from separate timepoints, which had a unreasonable rotation in the E14.5 sample. But during the 3D-to-3D alignment, PASTE2 had better quantification results (PCC and CI) and more reasonable alignment in the visualization results than PASTE and STAligner.

3. Regarding the sixth comment, it can be observed from the displayed accuracy that the majority of the mapping rates are less than 0.5. The authors may need to explain the reasons for the low accuracy or further demonstrate the feasibility of SANTO in cross-platform data mapping.

Thanks for your careful observation. The reasons of low accuracy of some cell types include: 1)

Since our method consider the global alignment between two slices, some major cell types (like R0 and R1) will dominate the alignment result, which means it will inevitably sacrifice the performance of alignment of minor cell types with chaotic spatial distributions (like R10 and R11). 2) The cell types of these two slices are annotated based on their own modalities separately as the ground truth. Therefore, these annotations are not consistent between the two slices and cannot match precisely, resulting in inevitable bias even after alignment. In this case, the accuracy does not range from 0 to 1. During our experiments, the highest accuracy for major cell type with clear distribution (like R0) is only 0.59. Thus, the range of accuracy for these cell types are determined by the consistency of original cell-type annotations of two slices.

Reviewer #3 (Remarks to the Author):

All my concerns have been addressed

We are very grateful to the three reviewers for their thoughtful and thorough comments, which definitely helped us improve our paper greatly. We have revised the paper following all of their comments. Below please find the point-by-point response to all the reviewers' comments.

=====

Reviewer #3 (Remarks to the Author):

All my concerns have been addressed

I am appreciated that you felt satisfied with our revision. Thanks for your comments on our manuscript.